# GROKKING AS A FIRST ORDER PHASE TRANSITION IN TWO LAYER NETWORKS

**Noa Rubin**[*]  Inbar Seroussi[†]  Zohar Ringel [‡]

## ABSTRACT

A key property of deep neural networks (DNNs) is their ability to learn new features during training. This intriguing aspect of deep learning stands out most clearly in recently reported Grokking phenomena. While mainly reflected as a sudden increase in test accuracy, Grokking is also believed to be a beyond lazy-learning/Gaussian Process (GP) phenomenon involving feature learning. Here we apply a recent development in the theory of feature learning, the adaptive kernel approach, to two teacher-student models with cubic-polynomial and modular addition teachers. We provide analytical predictions on feature learning and Grokking properties of these models and demonstrate a mapping between Grokking and the theory of phase transitions. We show that after Grokking, the state of the DNN is analogous to the mixed phase following a first-order phase transition. In this mixed phase, the DNN generates useful internal representations of the teacher that are sharply distinct from those before the transition.

## 1  INTRODUCTION

Feature learning is a process wherein useful representations are inferred from the data rather than being engineered. The success of deep learning is often attributed to this process. This is reflected, in part, by the performance gap between actual deep neural networks and their infinite-width Gaussian Process (GP) counterparts (Williams, 1996; Novak et al., 2018; Neal, 1996). It is also key to transfer learning applications (Weiss et al., 2016) and interpretability (Zeiler & Fergus, 2014; Chakraborty et al., 2017). Yet despite its importance, there is no consensus on how to measure or let alone classify feature learning effects.

Several recent results Li & Sompolinsky (2021); Ariosto et al. (2022) began shedding light on this matter. One line of work (adaptive kernel approaches) treats the covariance matrices of activation within each layer (kernels) as the key quantities undergoing feature learning. Feature learning would manifest as a deviation of these kernels from those of a random network and their adaptation to the task at hand. While providing a quantification of feature learning in quite generic settings, the equations governing these latent kernels are quite involved and may host a variety of learning phenomena. One such phenomenon (Seroussi et al., 2023), capable of providing a strong performance boost, is that of Gaussian Feature Learning (GFL) : A gradual process in which the covariance matrices of neuron pre-activations change during training so as to increase their fluctuations along label/target relevant directions. Remarkably, despite this smooth adaptation, the pre-activations' fluctuations, across width and training seeds, remain Gaussian. At the same time, the latent kernel itself develops notable spikes in the target direction, indicating feature learning.

Another phenomenon often associated with feature learning is Grokking. This abrupt phenomenon, first observed in large language models running on simple mathematical tasks, involves fast changes to the test accuracy following a longer period of constant and poor performance (Power et al., 2022). Though usually described as a time-dependent phenomenon, Grokking as a function of other parameters, it also occurs as a function of other parameters, such as sample size Power et al. (2022); Gromov (2023); Liu et al. (2022a). More broadly, DNNs behaviour as a function of time and dataset size

---

[*]Hebrew University, Racah Institute of Physics, Jerusalem, 9190401, Israel

[†]Department of Applied Mathematics, School of Mathematical Sciences, Tel Aviv University, Tel Aviv 69978, Israel

[‡]Hebrew University, Racah Institute of Physics, Jerusalem, 9190401, Israel

is often similar as reflected for instance the use of One Pass SGD You et al. (2014). Several authors provided quantitative explanations in the context of specific toy models wherein one can handcraft or reverse engineer the solution obtained by the network (Gromov, 2023; Nanda et al., 2023) or in suitably tailored perceptron models (Liu et al., 2022a; Žunkovič & Ilievski, 2022) where, however, representation learning is tricky to define. Given the aforementioned adaptive kernel approaches to deep learning, as well as the universality of Grokking across different DNNs and hyperparameters, it is natural to look for a more unifying picture of Grokking using a formalism that applies to generic deep networks.

In this work, we study Grokking as an equilibrium (or Bayesian) phenomenon driven by sample size, noise, or network width. Utilizing the aforementioned theoretical advancements, we show that Grokking in large-scale models can be classified and predicted through the mean field theory of phase transitions in physics (Landau & Lifshitz, 2013). Studying two different models, a teacher-student with cubic teacher and modular algebra arithmetic, we show the internal state of the DNN before Grokking is well described by GFL. In contrast, during Grokking, it is analogous to the mixed phase in the theory of first-order phase transitions, and the statistics of pre-activations are described by a mixture of Gaussian (GMFL). In this GMFL state, the latent kernels associated with the DNNs develop entirely new features that alter their sample complexity compared to standard infinite-width GP limits. After Grokking the weights are all specialized to the teacher. Besides providing a framework to classify feature learning effects, our approach provides analytically tractable and quantitative accurate predictions for the above two models.

Our main results are as follows:

- We establish a concrete mapping between the theory of first-order phase transitions, internal representations of DNNs, and Grokking for two non-linear DNN models each having two tunable layers.

- We identify three phases related to Grokking, one which is smoothly connected to the GP limit and two distinct phases involving different GP mixtures.

- For both our models, we simplify the task of learning high-dimensional representations to solving a non-linear equation involving either two (cubic teacher) or one (modular addition teacher) variables. Moreover, for the latter, we determine the location of the phase transition analytically.

- We flesh out a Grokking-based mechanism that can reduce the sample complexity compared to the GP limit.

**Prior works.** Phase transitions are ubiquitous in learning theory (e.g. Refs. Gardner & Derrida (1988); Seung et al. (1992); Györgyi (1990)), often in the context of replica-symmetry breaking. Connections between Grokking and phase transition were suggested by Nanda et al. (2023) but as far as analytic predictions go, prior work mainly focused on one trainable layer Žunkovič & Ilievski (2022); Arnaboldi et al. (2023) some suggesting those as an effective theory of representation learning Liu et al. (2022a). This can be further investigated by analyzing the loss landscape Liu et al. (2022b). Varma et al. (2023) suggest that the generalizing solution learned by the algorithm is more efficient but slower to learn than the memorizing one, using this interpretation they define regimes of semi-grokking, and ungrokking. The formalism of Refs. Arnaboldi et al. (2023); Saad & Solla (1995) can potentially be extended to online Grokking with two trainable layers, but would require reducing the large matrices involved. Phase transitions in representation learning have been studied in the context of bifurcation points in the information bottleneck approach (e.g. Tishby & Za-slavsky (2015)), nonetheless, the connection to deep learning remains qualitative. To the best of our knowledge, we provide a novel first-principals connection between grokking and phase transition *in representation learning*.

## 2 MODELS

### 2.1 NON-LINEAR TEACHER MODEL

Our first setting consists of a student Erf-network learning a single index non-linear teacher. The student is trained on a training set of size $n$, $\mathcal{D} = \{\boldsymbol{x}_\mu, y(\boldsymbol{x}_\mu)\}_{\mu=1}^n$ with MSE loss. In the following, bold symbol represents a vector. The input vector is $\boldsymbol{x}_\mu \in \mathbb{R}^d$ with iid Gaussian entries of variance

1. The target function $y$, is a scalar linear function of $\boldsymbol{x}$, with an orthogonal non-linear correction. Specifically, $y$ is given by-

$$y(\boldsymbol{x}) = \underbrace{\boldsymbol{w}^* \cdot \boldsymbol{x}}_{H_1(\boldsymbol{w}^* \cdot \boldsymbol{x})} + \epsilon \underbrace{\left( (\boldsymbol{w}^* \cdot \boldsymbol{x})^3 - 3 |\boldsymbol{w}^*|^2 \, \boldsymbol{w}^* \cdot \boldsymbol{x} \right)}_{H_3(\boldsymbol{w}^* \cdot \boldsymbol{x})}. \tag{1}$$

where $H_1, H_3$ are the first two odd Hermite polynomials, and $\boldsymbol{w}^* \in \mathbb{R}^d$ are the teacher weights. For simplicity we take here the norm of the teacher weights to be 1, but this has no qualitative effect on the theory as long as we require $|\boldsymbol{w}^*| \sim \mathcal{O}(1)$. We consider a fully connected non-linear student network with one hidden layer of width $N$ given by

$$f(\boldsymbol{x}) = \sum_{i=1}^{N} a_i \mathrm{erf}(\boldsymbol{w}_i \cdot \boldsymbol{x}). \tag{2}$$

where $\boldsymbol{w}_i \in \mathbb{R}^d$ for $i \in [1, N]$ are the students weights. Evidence that this model Groks can be found in App. (D).

## 2.2 GROKKING MODULAR ALGEBRA

Here we consider the setting of Ref. Gromov (2023), where the learning task is addition modulo $P$ where $P$ is prime. The network is trained on the following data set

$$\mathcal{D} = \{\boldsymbol{x}_{nm}, \boldsymbol{y}(\boldsymbol{x}_{nm}) \,|\, m, n \in \mathbb{Z}_P\} \tag{3}$$

where $\boldsymbol{x}_{nm} \in \mathbb{R}^{2P}$, is a vector such that it is zero in all its coordinates except in the coordinates $n$ and $P + m$ where it is 1 (a "two-hot vector"). The target function $\boldsymbol{y} \in \mathbb{R}^P$ is given by

$$y_p(\boldsymbol{x}_{nm}) = \delta_{p,(n+m) \bmod P} - 1/P, \tag{4}$$

where $\delta$ is the Kronecker delta and mod $P$ denotes the modulo operation which returns the remainder from the division by $P$. For the student model, we consider a two-layer deep neural network with a square activation function, given by

$$f_p(\boldsymbol{x}_{nm}) = \sum_{i=1}^{N} a_{pi}(\boldsymbol{w}_i \cdot \boldsymbol{x}_{mn})^2 \tag{5}$$

where $\boldsymbol{w}_c \in \mathbb{R}^{2P}$ for $c \in [1, N]$ are the students weights. For brevity, we denote $y_p(\boldsymbol{x}_{mn}) = y^p_{nm}$, and $f_p(\boldsymbol{x}_{nm}) = f^p_{mn}$.

## 2.3 TRAINING THE MODELS

In both cases, we consider networks that are trained with MSE loss to equilibrium using Langevin dynamics via algorithms such as Durmus & Moulines (2017); Neal et al. (2011). The continuum-time dynamics of the parameters are thus

$$\dot{\boldsymbol{\theta}}(t) = -\nabla_{\boldsymbol{\theta}} \left( \frac{\gamma}{2} \|\boldsymbol{\theta}(t)\|^2 + L(\boldsymbol{\theta}(t), \mathcal{D}) \right) + 2\sigma \boldsymbol{\xi}(t) \tag{6}$$

where $\boldsymbol{\theta}(t)$ is the vector of all network parameters in time $t$, $\gamma$ is the strength of the weight decay, $L$ is the loss function, the noise $\xi$ is given by $\langle \xi_i(t) \xi_j(t') \rangle = \delta_{ij} \delta(t - t')$ and $\sigma$ is the magnitude of the noise. We set the weight decay of the output layer so that with no data $a_i^2, a_{pc}^2$ both average to $\sigma_a^2/N$ under the equilibrium ensemble of fully trained networks. The input layer weights are required to have a covariance of $\sigma_w^2/d$ in the teacher-student model with $\sigma_w^2 = \mathcal{O}(1)$ and a covariance of 1 in the modular algebra model. Note that the covariance of the hidden layer is given by $\sigma^2/\gamma$. The posterior induced by the above training protocol coincides with that of Bayesian inference with a Gaussian prior on the weights defined by the above covariance and measurement noise $\sigma^2$ Naveh et al. (2021).

## 2.4 DERIVATION OVERVIEW

### 2.4.1 BRIEF INTRODUCTION TO MEAN FIELD THEORY PHASE TRANSITIONS

Phase transitions Landau & Lifshitz (2013); Tong (2011), such as the water-vapour transition, are ubiquitous in physics. They are marked by a singular behavior of some average observables as a function of a control parameter (say average density as a function of volume). As the laws of physics are typically smooth, phase transitions are inherently large-scale, or thermodynamic, phenomena loosely analogous to how the sum of many continuous functions may lead to a non-continuous one.

In a typical setting, the probability $p(x)$, of finding the system at a state $x$ can be approximately marginalized to track a single random variable called an order parameter ($\Phi$). As the latter is typically a sum of many variables (i.e. macroscopic), it tends to concentrate yielding a probability of the type $\log(p(\Phi)) \propto -d\tilde{S}(\Phi)$ where $d$ is a macroscopic scale (e.g. number of particles) and $\tilde{S}(\Phi)$ is some well behaved function that does not scale with $d$. Given this structure, the statistics of $\Phi$ can be analyzed using saddle point methods, specifically by Taylor expanding to second order around minima of $\tilde{S}$.

Phase transitions occur when two or more *global* minima of $\tilde{S}(\Phi)$ appear. First-order phase transitions occur when these minima are distinct before the phase transition and only cross in $\tilde{S}$ value at the transition. Notably, the effect of such crossing is drastic since, at large $d$, the observed behaviour (e.g. the average $\Phi$) would undergo a discontinuous change. Depending on the setup, such sharp change may be inconsistent as it would immediately change the constraints felt by the system. For instance, in the water-vapour transition, as one lowers volume the pressure on the vapour mounts. At some point, this makes a high-density minima of $\Phi$ as favourable as the low-density minima, signifying the appearance of water droplets. However, turning all vapor to droplets would create a drop in pressure making it unfavourable to form droplets. Instead, what is then observed is a mixture phase where as a function of the control parameter, a fraction of droplets forms so as to maintain two exactly degenerate minima of $\tilde{S}$. Lowering the volume further, a point is reached where all the vapour has turned into droplets and one is in the liquid phase.

In our analysis below, $\Phi$ will be a property of the weights in each neuron of the input layer, say their overlap with some given teacher weights. A high input dimension will be analogous to the large-scale limit, and the density loosely corresponds to the discrepancy in predictions. The phase transitions are marked by a new minima of $\tilde{S}(\Phi)$ which captures some feature of the teacher network useful in reducing the discrepancy. What we refer to as droplets corresponds to some input neurons attaining $\Phi$ values corresponding to the teacher feature while others fluctuate around the teacher agnostic minima. However the spatial notion associated with droplets is not relevant in this case as in the mean field theory of phase transitions.

### 2.4.2 ADAPTIVE KERNEL APPROACH AND ITS EXTENSION TO GAUSSIAN MIXTURES

Our main focus here is the posterior distribution of weights in the input layer ($p(\boldsymbol{w}_i)$) and posterior averaged predictions of the network ($f(\boldsymbol{x})$). Such posteriors are generally intractable for deep and/or non-linear networks. Hence, we turn to the approximation of Ref. Seroussi et al. (2023) where a mean-field decoupling between the read-out layer and the input layer is performed. This is exact in the limit of $N \to \infty$ and vanishing $\sigma_a^2$ (i.e. mean-field scaling). Following this, the posterior decouples to a product of two probabilities, a Gaussian probability for the read-out layer outputs and a generally non-Gaussian probability for the input layer weights. These two probabilities are coupled via two non-fluctuating quantities. The average kernel induced by the input layer and the discrepancy in predictions. As shown in Seroussi et al. (2023), specifically for a two-layer FCN, the resulting probability further decouples into iid probabilities over each neuron $p(\boldsymbol{w}_i)$. Below, we thus omit the neuron index $i$. The action ($-\log(p(\boldsymbol{w}_i))$ up to constant normalization factors) for each neuron's weights is then given by the following form

$$S[\boldsymbol{w}] = \frac{|\boldsymbol{w}|^2}{2\sigma_w^2} - \frac{\sigma_a^2}{2N} \sum_{\mu,\nu} \bar{\boldsymbol{t}}(\boldsymbol{x}_\mu)^T \bar{\boldsymbol{t}}(\boldsymbol{x}_\nu) \underbrace{\phi(\boldsymbol{w} \cdot \boldsymbol{x}_\mu) \phi(\boldsymbol{w} \cdot \boldsymbol{x}_\nu)}_{:=\sigma_a^{-2} \tilde{Q}_{\mu\nu}}, \tag{7}$$

where $\phi$ is the activation function and $\bar{t}$ is the discrepancy between the averaged network output and the target given by

$$\bar{t}(\boldsymbol{x}_\mu) = (\boldsymbol{y}(\boldsymbol{x}_\mu) - \overline{\boldsymbol{f}}(\boldsymbol{x}_\mu))/\sigma^2. \tag{8}$$

Notably $\bar{t}$ is not given but determined by solving the following two mean-field self-consistency equations

$$\overline{\boldsymbol{f}} = Q\left[Q + \sigma^2 I_n\right]^{-1}\boldsymbol{y} \tag{9}$$

where the kernel $Q$ is defined via

$$Q_{\mu\nu} = \sigma_a^2 \langle \phi(\boldsymbol{w} \cdot \boldsymbol{x}_\mu)\phi(\boldsymbol{w} \cdot \boldsymbol{x}_\nu) \rangle_{\mathcal{S}[\boldsymbol{w}]} \tag{10}$$

and $\langle...\rangle_{\mathcal{S}[\boldsymbol{w}]}$ denotes averaging over $\boldsymbol{w}$ with the probability implied by $\mathcal{S}[\boldsymbol{w}]$. In this work we use the equivalent kernel (EK) approximation, allowing the sums appearing in eqs. 7,9 to be replaced by integrals. This approximation washes out the generalization phenomena associated with Grokking, while capturing underlying feature learning mechanisms. As demonstrated in App. D, the feature learning effects hold also for finite datasets, in which a generalization gap can be observed. Theoretical corrections due to finite datasets can be made as shown in Cohen et al. (2021); Seroussi et al. (2023); Naveh & Ringel (2021), here we focus on the EK limit for simplicity.

Even if $\bar{t}$ is given, the remaining action is still non-linear. Ref. Seroussi et al. (2023) proceed by performing a variational Gaussian approximation on that action. Here we extend this approximation into a certain variational Gaussian *mixture* approximation. Specifically, we show that as one scales up $d, N, n$ in an appropriate manner, and following some decoupling arguments between, $\mathcal{S}[\boldsymbol{w}]$ has the form $d\hat{\mathcal{S}}[\Phi(\boldsymbol{w})]$ where $d \gg 1$ and $\hat{\mathcal{S}}[\Phi(\boldsymbol{w})]$ has $O(1)$ coefficients and $\Phi(\boldsymbol{w})$ is some linear combination of the weights. This allows us to treat the integration underlying $\langle...\rangle_{\mathcal{S}[\boldsymbol{w}]}$ within a saddle-point approximation. Notably when more than one Global saddle appears the saddle-point treatment corresponds to $p(\boldsymbol{w})$ having a Gaussian mixture form.

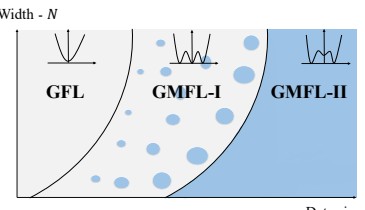

Figure 1: Schematic phase diagram of learning phases. The inset plots in the different phases correspond to the normalized negative log posterior ($\hat{\mathcal{S}}$). A transition between the different phases can be achieved by varying either $n$ or $N$.

The three phases of learning described in the introduction correspond to the following behavior of the saddles (see also illustration in Fig. 1). At first, a single saddle centered around $\Phi(\boldsymbol{w}) = 0$ exists and weights fluctuate in a Gaussian manner around these minima, as in GFL. In the next phase, the distribution is comprised of a weighted average of this zero-saddle and other $|\Phi(\boldsymbol{w})| > 0$ saddles. This marks a new learning ability of the network and hence the beginning of Grokking. We name this mixture phase the first Gaussian Mixture Feature Learning phase (GMFL-I). This phase corresponds to the mixed ("droplets") phase. In this phase, picking a random neuron, there is a finite chance it is in the GFL phase and hence fluctuates around the target agnostic minima at $\boldsymbol{w} = 0$. Similarly, there is a finite chance it fluctuates around one of the non-trivial saddles. At some later point, after more data has been presented, only the saddles with $|\boldsymbol{w}| > 0$ dominate the average (GMFL-II). The phase phenomenology is shared by both models, however, the details of the new saddles that appear, as well as the decoupling scheme between the different components of $\boldsymbol{w}$ differ between both models. We next turn our attention to these details.

## 3 RESULTS

### 3.1 NON-LINEAR TEACHER STUDENT MODEL

**Scaling setup and effective interaction.** Consider the following two scaling variables $(\alpha, \beta)$ which we would soon take to infinity together and consider scaling up the microscopic parameters in the following manner

$$N \to \beta N \quad d \to \sqrt{\beta}d \quad \sigma_a^2 \to \sigma_a^2/\sqrt{\beta} \ , \ \sigma^2 \to \frac{\alpha}{\beta}\sigma^2 \quad n \to \alpha n \tag{11}$$

where we comment that $\alpha$ can be seen as a continuum approximation allowing us to replace data summation with integrals over the data measure and $\beta$ is a combination of mean-field-type scaling Mei et al. (2018) together with a thermodynamic/saddle-point limit. Notably the following combination ($u$) of hyper-parameter $u = \frac{n^2\sigma_a^2}{\sigma^4 dN}$, which we refer to as the effective interaction, is invariant under both $\alpha$ and $\beta$.

**Claim I. Two relevant discrepancy modes before the transition.** For $\beta \to \infty$ and $\sqrt{\beta}/\alpha \to 0$, and $u \leq u_c(\epsilon)$ ($\approx 30.2$ for $\epsilon = -0.3$) the discrepancy takes the following form

$$\sigma^2 \bar{t}(\boldsymbol{x}) = bH_1(\boldsymbol{x}) + cH_3(\boldsymbol{x}) \tag{12}$$

where $b, c \in \mathbb{R}$ are some $O(1)$ constant coefficients. We comment that $ub^2$ is proportional to the emergent scale of Ref. Seroussi et al. (2023). For further detail see App. A.2.

**Claim II. One-dimensional posterior weight distribution.** In the same scaling limit, $u \leq u_c$ the negative log probability (action in physics terminology) of weights along the teacher direction, decouples from the rest of the $\boldsymbol{w}$ modes, and takes the following form

$$\mathcal{S}[\boldsymbol{w} \cdot \boldsymbol{w}^*] = d\left(\frac{(\boldsymbol{w} \cdot \boldsymbol{w}^*)^2}{2\sigma_w^2} - \frac{2n^2\sigma_a^2}{\pi\sigma^4 dN}\frac{(\boldsymbol{w} \cdot \boldsymbol{w}^*)^2}{1 + 2\left(\sigma_w^2 + (\boldsymbol{w} \cdot \boldsymbol{w}^*)^2\right)}\left(b - \frac{2c(\boldsymbol{w} \cdot \boldsymbol{w}^*)^2}{1 + 2\left(\sigma_w^2 + (\boldsymbol{w} \cdot \boldsymbol{w}^*)^2\right)}\right)^2\right) \tag{13}$$

Notably, this expression reduces the high-dimensional network posterior into a scalar probability involving only the relevant order parameter ($\Phi = \boldsymbol{w} \cdot \boldsymbol{w}^*$). We further note that all the expressions in the brackets are invariant under $\alpha$ and $\beta$. Heuristically, this will also be the action for a small $\Delta u$ after the phase transition since the resulting corrections to the discrepancy have a parametrically small effect. For further detail see App. A.2.

**Claim III. Exactness of Gaussian Mixture Approximation.** In the same scaling limit, the probability described by the above action is exactly a mixture of Gaussians each centred around a global minimum of $\mathcal{S}$.

**Claim IV. First-order phase transition.** For $u < u_c$ the only saddle is that at $\Phi = 0$. Exactly at $u = u_c$ three saddles appear two of which are roughly at $\Phi = \pm|w_*|^2 = \pm 1$. For some finite interval $u \in [u_c, u_c + \Delta u]$, these saddles stay degenerate in $\hat{\mathcal{S}}$ value.

### 3.1.1 Teacher-Student Experimental Results

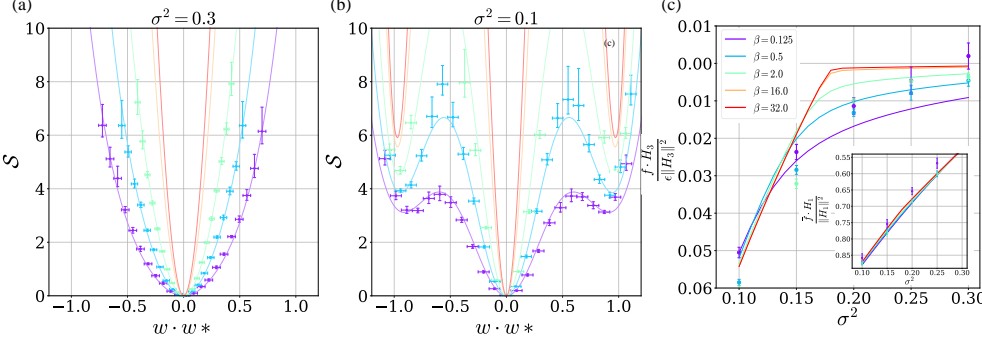

**Figure 2: GFL to GMFL-I Theory and Experiment in the teacher-student model.** Panel (a) and (b) show the negative log posterior before and after the phase transition induced by varying the noise $\sigma^2$. A good match with the experimental values (crosses) and theory (solid lines) for rarer and rarer events is obtained as we scale up the model according to Table 1. (colour coding). Turning to network outputs, panel (c) shows the expected phase transition in learning the cubic teacher component and the inset shows the discrepancy in the linear teacher component. As our analytics holds before and at the phase transitions, discrepancies in $\bar{f} \cdot H_3$ close to the transition are an expected finite-size effect made pronounced by its low absolute value.

To validate our theoretical approach, we trained an ensemble of 200 DNNs for different $\alpha = \beta$ values using our Langevin dynamics at sufficiently low learning rates and for a sufficiently long

time to ensure equilibration. Our initial $N, d, \sigma^2, \sigma_a^2$ (i.e. at $\alpha = \beta = 1$) were: $N = 700, n = 3000, d = 150, \sigma_w^2 = 0.5, \sigma_a^2 = 8/N$ and we took $\epsilon = -0.3$. Our training ensemble consisted of both different initialization seeds and different data-draw seeds. These, together with the neuron indices associated $N$, provided $200N$ draws from $\boldsymbol{w}_* \cdot \boldsymbol{w}$ used for estimated $p(\boldsymbol{w})$. The discrepancy was estimated using a dot product of the outputs with $H_{1/3}(x)$ sampled over 3000 test points and then averaged over the 50 seeds.

Our experimental results are given in Fig. 2. Panel (c) shows how the linear (inset) and cubic target components learned, as a function of $\sigma^2$. Notably, reducing $\sigma^2$ is similar to increasing $n$, hence one expects more feature learning at lower $\sigma^2$. As we increase $\alpha = \beta$ (see color coding) a sharp phase transition develops around $\sigma^2 = 0.18$ where the cubic component begins its approach to the teacher's value ($\epsilon$). Panels (a) and (b) track $-\log(p(\boldsymbol{w} \cdot \boldsymbol{w}_*))$ at two points before and after the phase transition, respectively. As $\alpha = \beta$ increases, the theory predicts a finite probability for $\Phi(\boldsymbol{w}) = \boldsymbol{w} \cdot \boldsymbol{w}_* \approx 1$. This means that picking a random neuron has a finite chance of fluctuating around the teacher-aware minima. Such neurons are what we refer to as our droplets. All graphs show a good agreement with theory as one scales up $\alpha = \beta$ before the transition. The remaining discrepancies are attributed to finite size (i.e. finite $\alpha, \beta$) effects which, as expected, become more noticeable near the transition.

The above experiment also points to some potentially powerful complexity aspects of feature learning. Notably, an FCN NNGP kernel induces a uniform prior over cubic polynomials (i.e. all $l = 3$ hyper-spherical harmonics). As there are $n = O(d^3)$ of those, it requires $n = 0.5e + 6$ datapoints to learn such target components in our setting (see also Ref. Cohen et al. (2021)). Here this occurs two orders of magnitudes earlier ($n = 3000$). This occurs because the complex prior induced by a *finite* DNN learns the features from the readily accessible linear components of the target and applies them to the non-linear ones. Proving that such "assisted learning" of complex features changes the sample complexity compared to NNGPs requires further work. In App. A.1 we provide an analytical argument in support of this.

Table 1: Scaling laws

| Model | Width | Input-dimension | Data size | Noise strength | Weight decay |
|---|---|---|---|---|---|
| Polynomial regression | $N \to \beta N$ | $d \to \sqrt{\beta} d$ | $n \to \alpha n$ | $\sigma^2 \to \frac{\alpha}{\beta}\sigma^2$ | $\sigma_a^2 \to \sigma_a^2/\sqrt{\beta}$ |
| Modular Theory | $N \to \beta^2 N$ | $P \to \sqrt{\beta} P$ | $P^2 \to \beta P^2$ | $\sigma^2 \to \sigma^2/\beta$ | $\sigma_a^2 \to \sigma_a^2/\beta$ |

### 3.2 MODULAR ALGEBRA THEORY

**Scaling setup and effective interaction.**

Similar to the polynomial regression problem, we consider a scaling variable ($\beta$) which we later take to infinity together, and consider scaling up the microscopic parameters. The precise scaling is given in Table 1.

$$N \to \beta^2 N \quad P \to \sqrt{\beta} P \quad \sigma_a^2 \to \sigma_a^2/\beta \quad \sigma^2 \to \sigma^2/\beta \tag{14}$$

Note that, here, we do not need additional scale for the continuum limit of the dataset, since the continuum limit is taken by considering all combinations of data points. As before, $\beta$ is a combination of mean-field scaling together with a thermodynamic/saddle-point limit. Notably, the following combination ($u$) of hyperparameter $u = \frac{2\sigma_a^2 P^2}{N\sigma^4}$, which we refer to as the effective interaction, is invariant under $\beta$.

**Problem Symmetries**

The following symmetries of $S[\boldsymbol{w}]$ (which is also a function of $\bar{\boldsymbol{t}}$) help us decouple the posterior probability distribution into its Fourier modes and simplify the problem considerably:

**I.** Taking $[n, m] \to [(n + q) \bmod P, (m + q') \bmod P]$, and $f_p \to f_{(p+q+q') \bmod P}$ with $q, q' \in \mathbb{Z}_P$

**II.** Taking $[n, m] \to [qn \bmod P, qm \bmod P]$ and $f_p \to f_{qp \bmod P}$ for $q \in \mathbb{Z}_P$ but different than zero.

**Claim I. Single discrepancy mode.** Several outcomes of these symmetries are shown in App. (B.1). First, we find that the adaptive GP kernel $Q$ (given explicitly in Eq. 21) is diagonal in the

basis $\phi_{k,k'}(x_{n,m}) = P^{-1}e^{2\pi i(kn+k'm)/P}$ , where $k, k' \in \{0, 1, ..., P-1\}$. Considering eigenvalues, the second symmetry implies that $\phi_{k,k'}$ would be degenerate with $\phi_{ck,ck'}$. For prime, $P$ this implies, in particular, that all $\phi_{k,k}$ eigenvectors with $k > 0$ have the same eigenvalue ($\lambda$). Notably, the target itself is spanned by this degenerate subspace specifically

$$y_{nm}^p = P^{-1} \sum_{k=1}^{P-1} e^{-i2\pi kp/P} e^{i2\pi k(n+m)/P} = \left[ \sum_{k=1}^{P-1} e^{-i2\pi kp/P} \phi_{k,k}(x_{n,m}) \right] \tag{15}$$

As a result, one finds that the target is always an eigenvector of the kernel and $\sigma^2 \bar{t}_{mn}^p = a y_{nm}^p$ where $a \in \mathbb{R}$. Thus there is only one mode of the discrepancy which is aligned with the target.

**Claim II. Decoupled two-dimensional posterior weight distribution for each Fourier mode.** We decouple all the different fluctuating modes by utilizing again the symmetries of the problem and making a judicial choice of the non-linear weight decay term ($\Gamma$). To this end, we define the following Fourier transformed weight variables $(w_k, v_k)$: $w_n = \sum_{k=0}^{P-1} w_k e^{-\frac{2\pi ikn}{P}}$, $w_m = \sum_{k=0}^{P-1} v_k e^{-\frac{2\pi ikm}{P}}$ which when placed into action yields

$$\mathcal{S}[\hat{\boldsymbol{w}}] = P \left[ \left( \frac{1}{2} \sum_{k=0}^{P-1} w_k w_{-k} + \frac{1}{2} \sum_{k=0}^{P-1} v_k v_{-k} \right) - \frac{2\sigma_a^2 a^2 P^2}{N\sigma^4} \sum_{k=1}^{P-1} w_k w_{-k} v_k v_{-k} \right] + \Gamma[\boldsymbol{w}] \tag{16}$$

(see App. B.2) where apart from the non-linear weight-decay term, all different $k$ modes have been decoupled. For simplicity, we next choose, $\Gamma[\boldsymbol{w}] = \sum_k P\frac{\gamma}{6}\left( (w_k w_{-k})^3 + (v_k v_{-k})^3 \right)$. Here there are technically two order parameters- $\Phi = w_k w_{-k}, \Psi = v_k v_{-k}$, but from the symmetry of the action, we obtain that the saddles occur only at points where $\Phi = \Psi$ . We comment that the analysis of a more natural weight decay terms such as $\sum_n w_n^6 + w_m^6$, using a certain GP mixture ansatz of $p(w_i)$ as an approximation, we obtained similar qualitative results.

**Claim III. Exactness of Gaussian Mixture Approximation.** Following the presence of a large factor of $P$ in front of the action, the above non-linear action, per $k-$mode, can be analyzed using standard saddle point treatment. Namely, treating $Q$ as a function of $a$, and with it $\lambda$, can be evaluated through a saddle-point approximate on the probability associated with this action. In the limit of $\beta \to \infty$, we obtain that this approximation is exact. Following, this $a$ can be calculated using the GPR expression $-\frac{\sigma^2}{\lambda + \sigma^2}$, and in this limit the value of $\lambda$ can be computed exactly. Demanding this latter value of $a$ matches the one in the action results in an equation for $a$.

**Claim IV. First-order phase transition.** Since the quadratic term is constant in the scaled action, as long as no other saddles become degenerate (in action value) with the $\Phi = \Psi = 0$ saddle, the saddle-point treatment truncates the action at this first term. By increasing $u$ the quartic term becomes more negative, and hence a first-order symmetry-breaking transition must occur at some critical value of $u$. Past this point, $a$ begins to diminish. If it will diminish too rapidly, the feedback on the action will be such that it is no longer preferential for $a$ to diminish, and thus a probability distribution will have two degenerate minima at a zero and non zero value of $\Phi$ representing the GFML-I phase. Further increasing $u$ will break the degeneracy resulting in the global minimum of the log posterior distribution being non trivial. Notably $a$ measures the test-RMSE here, thus the fact that it remains constant and suddenly begins to diminish can also be understood as Grokking.

### 3.2.1 MODULAR ALGEBRA NUMERICAL SIMULATIONS

Solving the implied equation for $a$ numerically yields the full phase diagram here (see App. B.3 for technical details of solution) supporting the assumptions in **Claim IV**. Fig. (3) plots the negative-log-probability of weights for an arbitrary $k$ taking $\Phi = \Psi$ for simplicity, since at the global minima this is anyways true. Here we increase $u$ by decreasing $\sigma^2$ and find the action by solving the equation for $a(\sigma^2)$ numerically. The $\Phi \neq 0$ saddles are exponentially suppressed in $P$, yet nonetheless become more probable as $\sigma^2$ decreases. At around, $\sigma^2 \approx 0.227$ they come within $\mathcal{O}(1)$ of the saddle at $S_k(\Phi = 0, \Psi = 0)$ ($\mathcal{O}(1/P)$ in the plot, given the scaled y-axis). This marks the beginning of the mixed phase (GMFL-I), wherein all action minima contribute in a non-negligible manner. Further, in this phase, decreasing $\sigma^2$ does not change the height of the $\Phi \neq 0$ minima (see inset) in any appreciable manner. Had we zoomed in further, we would see a very minor change to these saddle's

height throughout the mixed phase, as they go from being $\mathcal{O}(1)$ above the $\Phi = 0$ saddle to $\mathcal{O}(1)$ below that saddle at $\sigma^2 \approx 0.175$, this is shown in the inset graph in Fig. (3). This latter point marks the beginning of the GMFL-II phase, where it is the contribution of the minimum at $\Phi = 0$ which becomes exponentially suppressed in $P$. Notably $a$, which measures here the test-RSME, goes from $-1$ at the beginning of GMFL-I to $-0.7$ at its end. Over this small interval of $\sigma^2$ we observe a $30\%$ reduction in the magnitude of the discrepancy which can be thought of as a manifestation of Grokking.

Our analytical results are consistent with the experiments carried out in Ref. Gromov (2023). Indeed, as we enter GMFL-I, weights sampled near the $\Phi, \Psi \neq 0$ saddle, correspond to the cosine expressions for $W_k^{(1)}$ of that work. As our formalism marginalizes over readout layer weights, the phase constraint suggested in their Eq. (12) becomes irrelevant, and both viewpoints retrieve the freedom of choosing cosine phases. In our case, this stems from the $U(1) \times U(1)$ complex-phase freedom in our choice of saddles at $\Phi, \Psi > 0$.

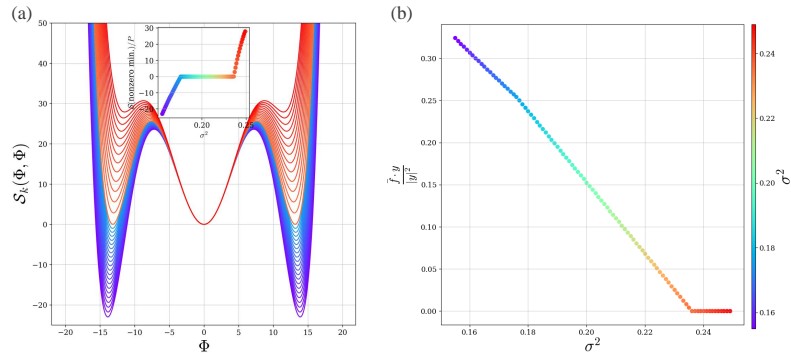

Figure 3: **GFL to GMFL-I to GMFL-II** (a) Probability distribution of weights as predicted by our approach. The GFL phase is represented by the red graphs, where the minimum of the action at zero (shared also by the GP limit) dominates the probability distribution. The GMFL-I phase can be seen in the inset graph, and the final GMFL-II phase is shown in purple.(b) The target component of the average network output. Here singularities can be observed at the $\sigma^2$ values where the phase transitions occur. The Parameters taken in this calculation are: $N = 1000, P = 401, \sigma_a^2 = 0.002/N, \gamma = 0.0001$

## 4 DISCUSSION

In this work, we studied two different models exhibiting forms of Grokking and representation/feature learning. We argue that Grokking is a result of a first order phase transition. To analyze this analytically, we extended the approach of Seroussi et al. (2023) to include mixtures of Gaussian. The resulting framework led to concrete analytical predictions for rich feature learning effects exposing, in particular, several phases of learning (GFL,GMFL-I,GMFL-II) in the thermodynamic/large-scale limit. Our results also suggest that feature learning in finite FCNs with mean-field scaling can change the sample complexity compared to the associated NNGP. Certainly, these describe very different behavior compared to the recently explored kernel-scaling Li & Sompolinsky (2021); Ariosto et al. (2022) approach, wherein feature learning amounts to a multiplicative factor in front of the output kernel. A potential source of difference here is their use of standard scaling, however, this remains to be explored.

As our results utilize a rather general formalism Seroussi et al. (2023), we believe they generalize to deep networks and varying architecture. As such, they invite further examination of feature learning in the wild from the prism of the latent kernel adaptation . Such efforts may provide, for instance, potential measures of when a model is close to Grokking by tracking outliers in the weight or pre-activation distributions along dominant kernel eigenvectors. As latent kernels essentially provide a spectral decomposition of neuron variance, they may help place empirical observations on neuron sensitivity and interpretability Zeiler & Fergus (2014) on firmer analytical grounds. Finally, they suggest novel ways of pruning and regulating networks by removing low-lying latent kernel eigenvalues from internal representations.

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

# SUPPLEMENTARY MATERIAL

## A    TEACHER-STUDENT MODEL

### A.1    SCALING LIMIT OF TEACHER-STUDENT MODEL AND SAMPLE COMPLEXITY ASPECTS

Here we detail the scaling limits in which we believe our results become exact. To this end, we first point to the factors controlling the three approximations we employ and discuss scaling limits in which these factors all vanish.

Our first approximation is the mean-field decoupling between the read-out layer and the input layer. Specifically we follow Mei et al. (2018); Seroussi et al. (2023); Bordelon & Pehlevan (2022), and introduce a mean-field scaling parameter $\chi$ define $\sigma_a^2 = \tilde{\sigma}_a^2/\chi$ (i.e. the microscopic variance of each readout layer weight is $\tilde{\sigma}_a^2/(\chi N)$ and $\sigma^2 = \tilde{\sigma}^2/\chi$ (i.e. the noise variance on the gradients is $\sigma^2$). Notably, since $b, c \propto \sigma^2/[\sigma^2 + K]$ and $K \propto \sigma_a^2$, we finds that $\chi$ has no effect on $b$ or $c$. Examining the action, this yields an additional constant factor of $\chi$ in front of the non-linear term, as expected since mean-field scaling increases feature learning. Taking $\chi \gg 1$ controls the accuracy of our mean-field decoupling between the read-out and input layer.

Our second approximation is the continuum limit, where we trade summation over data-points with integrals. This approximation is closely related to the EK approximation Rasmussen & Williams (2005); Cohen et al. (2021). Taking $n$ to be large while keeping $n/(\tilde{\sigma}^2) \equiv n_{\text{eff}}$ fixed, corrections to this approximation scale as $1/\tilde{\sigma}^2$. Notably taking large $\tilde{\sigma}^2$ while keeping $n_{\text{eff}}$ fix does not change $b, c$ or the action of the weights as the latter contains only the combination $n/\tilde{\sigma}^2$.

The third approximation we carry is the saddle-point approximation or Gaussian-mixture approximation for $p(\boldsymbol{w})$. Examining our action

$$S\left[\boldsymbol{w}\right] = d\left(\frac{|\boldsymbol{w}|^2}{2\sigma_w^2} - \frac{2\chi n_{\text{eff}}^2 \tilde{\sigma}_a^2}{\pi dN}\frac{(\boldsymbol{w}\cdot\boldsymbol{w}^*)^2}{1+2|\boldsymbol{w}|^2}\left(b[n_{\text{eff}}/d] - \frac{2c[\epsilon]\,(\boldsymbol{w}\cdot\boldsymbol{w}^*)^2}{1+2|\boldsymbol{w}|^2}\right)^2\right) \tag{1}$$

where we further indicated the dependencies of $b$ and $c$ which arise through GPR, $d$ comes out as controlling the saddle point approximation, provided the other coefficients in the action do not scale down with $d$.

We thus find that the following family of scaling limits controlled by two scaling parameters ($\alpha, \beta \to \infty$) ensures $S[\boldsymbol{w}]$ is exact and its saddle points dominate $\boldsymbol{w}$

$$N \to \beta N \tag{2}$$
$$d \to \sqrt{\beta}d$$
$$\chi \to \sqrt{\beta}\chi$$
$$\tilde{\sigma}^2 \to \frac{\alpha}{\sqrt{\beta}}\tilde{\sigma}^2 \quad \sigma^2 \to \frac{\alpha}{\beta}\sigma^2$$
$$n \to n\alpha$$

Notably increasing $\alpha$ makes the continuum approximation exact Cohen et al. (2021) whereas increasing $\beta$ makes both the saddle point and the mean-field decoupling exact. Furthermore, it can be verified that both $\alpha$ and $\beta$ have no effect on $\tilde{S}$ and in particular keep the coefficient in front of the non-linear term invariant. The only conflict between these two limits is in $\tilde{\sigma}^2$ hence we require $\alpha^2 > \beta$, but otherwise one is free to scale these up differently.

We conjecture that the teacher-student model has a better sample complexity than the infinite-width network described by the GP limit. Namely, the GP limit is able to learn the cubic target at, $n \propto d^3$ whereas the former can reach good accuracy at $n \propto d$. Replacing $n$ with the above, $n_{\text{eff}}$ the latter is implied by the linear scaling of and $d, n_{\text{eff}}$ together with our analytical results, which show that the cubic component is being learned. In general, reduction of complexity has been seen in the context of Gradient descent in Bietti et al. (2022); Sarao Mannelli et al. (2020); Gamarnik et al. (2019) as well as in online learning up to logarithmic factors Arous et al. (2021); Mousavi-Hosseini et al. (2022); Tan & Vershynin (2019).

Establishing this more rigorously requires showing that $n$ itself, and not $n/\tilde{\sigma}^2$ can be kept proportional to $d$. Alternatively stated, that $\alpha$ can scale as $\sqrt{\beta}$ while introducing corrections to the theory proportional to $\sqrt{\beta}/\alpha$, which one can then take to be as small as desired. This scenario seems to be consistent with the corrections to EK derived in Ref. Cohen et al. (2021), which when evaluated in simple settings (random data on a hypersphere and a rotationally invariant kernel) indeed show that $\sqrt{\beta}/\alpha$ controls corrections to EK.

## A.2    CONTINUUM ACTION AND MODE DECOUPLING

Our first step here consists of removing the randomness induced by the specific choice of training set. To this end, we consider taking a large $n$ limit while keeping $n/\sigma^2$ fixed. This has several merits. At large, $n$ it is natural to replace summations in Eq. (7) by integrals, which can be evaluated directly. Furthermore, in this limit, one obtains an analytical handle on Gaussian processes regression via the equivalence kernel (EK) approximation Rasmussen & Williams (2005); Cohen et al. (2021). Notably, previous works have found reasonable convergence to this approximation, in similar scenarios to ours, for values as low as $n = 100$ and $\sigma^2 = 0.1$ Cohen et al. (2021). A key element in the EK approximation is the spectrum of the kernel defined by

$$\int d\mu_z Q(\boldsymbol{x}, \boldsymbol{z})\phi_\lambda(\boldsymbol{z}) = \lambda\phi_\lambda(\boldsymbol{x}) \tag{3}$$

where $\mu_z$ is the dataset measure, which corresponds here to i.i.d. standard Gaussians. Denoting by $g_\lambda$ the spectral decomposition of the target function on $\phi_\lambda(\boldsymbol{x})$, the GPR predictions (Eq. 9) are given by

$$\bar{f}(\boldsymbol{x}) = \sum_\lambda \frac{\lambda}{\lambda + \sigma^2/n} g_\lambda \phi_\lambda(\boldsymbol{x}) \tag{4}$$

where $\frac{\lambda}{\lambda + \sigma^2/n}$ are the learnability factors. Notably, $n$ only enters through the combination, $n/\sigma^2$ which we treat as the effective amount of data. Hence, we can control the amount of learning in two equivalent ways, by either providing more data points or by reducing the noise.

The resulting limit also enables us to identify the two relevant directions in function space, being $H_1(\boldsymbol{x})$ and $H_3(\boldsymbol{x})$. Specifically we argue that at large $d$ the operator $Q(\boldsymbol{x}, \boldsymbol{y})$ is block-diagonal within the space spanned by $H_1(\boldsymbol{x})$ and $H_3(\boldsymbol{x})$. Since it acts on a target having support only on $H_1(\boldsymbol{x}), H_3(\boldsymbol{x})$, this limits $\bar{f}(\boldsymbol{x})$ to this space and hence also $\bar{t}(\boldsymbol{x}) = [y(\boldsymbol{x}) - \bar{f}(\boldsymbol{x})]/\sigma^2$. Our claim is shown to be self-consistent in App. A namely, that making this assumption results in $Q$ and $\bar{t}$ which obey the same assumption. Operationally, we thus take the ansatz

$$\sigma^2 \bar{t}(\boldsymbol{x}) = bH_1(\boldsymbol{x}) + cH_3(\boldsymbol{x}) \tag{5}$$

where $b, c \in \mathbb{R}$ are some unknown constant coefficients. Later, we derive self-consistent equations determining these constants. Given the above ansatz, we turn to the action in Eq. 7, take the continuum approximation where $\sum_{\mu=1}^n \to n \int d\mu_y$ and find that the integral appearing in the action can be solved exactly- $\bar{t}(\boldsymbol{x}) = bH_1(\boldsymbol{w}^* \cdot \boldsymbol{x}) + cH_3(\boldsymbol{w}^* \cdot \boldsymbol{x})$, we can compute the action in the continuum limit-

$$\mathcal{S} \equiv \frac{d|\boldsymbol{w}|^2}{2\sigma_w^2} - \frac{n^2\sigma_a^2}{2N\sigma^4} \left( \left(b - 3c|\boldsymbol{w}^*|^2\right) \underbrace{\int d\mu_x (\boldsymbol{w}^* \cdot \boldsymbol{x})\phi(\boldsymbol{w} \cdot \boldsymbol{x})}_{I_0} + c \underbrace{\int d\mu_x (\boldsymbol{w}^* \cdot \boldsymbol{x})^3 \phi(\boldsymbol{w} \cdot \boldsymbol{x})}_{I_1} \right)^2 \tag{6}$$

$$= d \left( \frac{|\boldsymbol{w}|^2}{2\sigma_w^2} - \frac{2n^2\sigma_a^2}{\pi d N\sigma^4} \frac{(\boldsymbol{w} \cdot \boldsymbol{w}^*)^2}{1 + 2|\boldsymbol{w}|^2} \left(b - \frac{2c(\boldsymbol{w} \cdot \boldsymbol{w}^*)^2}{1 + 2|\boldsymbol{w}|^2}\right)^2 \right)$$

we the integrals $I_0, I_1$ are computed in App. A.4.

In principle, one could perform saddle point analysis in two variables. A simplification can be made by noting that the dominant fluctuations are in the direction of $\boldsymbol{w}^*$. We are therefore projecting

$\boldsymbol{w}$ into the subspace of $\boldsymbol{w}^*$, such that, $\boldsymbol{w} = P_{\boldsymbol{w}^*}\boldsymbol{w} + P_{\boldsymbol{w}^*}^{\perp}\boldsymbol{w}$, with $P_{\boldsymbol{w}^*} = \frac{1}{|\boldsymbol{w}^*|^2}\boldsymbol{w}^*(\boldsymbol{w}^*)^T$, and $P_{\boldsymbol{w}^*} = I_d - P_{\boldsymbol{w}^*}^{\perp}$, Denote $\alpha = \boldsymbol{w}^* \cdot \boldsymbol{w}$ and $\alpha_{\perp} = |P_{\boldsymbol{w}^*}^{\perp}\boldsymbol{w}|$ the action is given by

$$\mathcal{S}[\boldsymbol{w}] = \frac{d}{2\sigma_w^2}\alpha^2 + \frac{d}{2\sigma_w^2}\alpha_{\perp}^2 - \frac{4n^2\sigma_a^2}{\pi 2N\sigma^4}\frac{|\boldsymbol{w}^*|^2\alpha^2}{1+2(\alpha^2+\alpha_{\perp}^2)}\left(b - \frac{2c|\boldsymbol{w}^*|^2\alpha^2}{1+2(\alpha^2+\alpha_{\perp}^2)}\right)^2, \quad (7)$$

By concentration of the norm, we replace $\alpha_{\perp}$ by its average $\sigma_w^2$ yields the following action-

$$\tilde{\mathcal{S}}[\boldsymbol{w}\cdot\boldsymbol{w}^*] = d\left(\frac{(\boldsymbol{w}\cdot\boldsymbol{w}^*)^2}{2\sigma_w^2} - \frac{2n^2\sigma_a^2}{\pi\sigma^4 dN}\frac{(\boldsymbol{w}\cdot\boldsymbol{w}^*)^2}{1+2\left(\sigma_w^2+(\boldsymbol{w}\cdot\boldsymbol{w}^*)^2\right)}\left(b - \frac{2c(\boldsymbol{w}\cdot\boldsymbol{w}^*)^2}{1+2\left(\sigma_w^2+(\boldsymbol{w}\cdot\boldsymbol{w}^*)^2\right)}\right)^2\right) \quad (8)$$

## A.3 Solving the equations for $b, c$

Since the kernel does not mix polynomials of higher order, we can write the GPR expression for the mean network output in the function space spanned by the components of the target. Namely, any function in this two-dimensional function space is given by a linear combination of $\hat{H}_1(\boldsymbol{x}) = H_1(\boldsymbol{x}), \hat{H}_3(\boldsymbol{x}) = \frac{1}{\sqrt{6}}H_3(\boldsymbol{x})$. The factor of $\frac{1}{\sqrt{6}}$ results from the requirement that the basis of this space will be normal. In this basis, the discrepancy and the target is given by,

$$\bar{\boldsymbol{t}} = \begin{bmatrix} b \\ \sqrt{6}c \end{bmatrix}, \boldsymbol{y} = \begin{bmatrix} 1 \\ \sqrt{6}\epsilon \end{bmatrix} \quad (9)$$

In this basis, the two dimensional kernel $\tilde{Q}$ is given by

$$\tilde{Q} = \begin{bmatrix} \int d\mu_{\boldsymbol{x}}d\mu_{\boldsymbol{z}}\hat{H}_1(\boldsymbol{x})Q(\boldsymbol{x},\boldsymbol{z})\hat{H}_1(\boldsymbol{z}) & \int d\mu_{\boldsymbol{x}}d\mu_{\boldsymbol{z}}\hat{H}_3(\boldsymbol{x})Q(\boldsymbol{x},\boldsymbol{z})\hat{H}_1(\boldsymbol{z}) \\ \int d\mu_{\boldsymbol{x}}d\mu_{\boldsymbol{z}}\hat{H}_1(\boldsymbol{x})Q(\boldsymbol{x},\boldsymbol{z})\hat{H}_3(\boldsymbol{z}) & \int d\mu_{\boldsymbol{x}}d\mu_{\boldsymbol{z}}\hat{H}_3(\boldsymbol{x})Q(\boldsymbol{x},\boldsymbol{z})\hat{H}_3(\boldsymbol{z}) \end{bmatrix} \quad (10)$$

where $\mu_{\boldsymbol{x}}$ is the standard Gaussian measure, $Q$ is the mean continuum limit kernel defined as

$$Q(\boldsymbol{x},\boldsymbol{z}) = \sigma_a^2\int\frac{e^{-\mathcal{S}(\boldsymbol{w};b,c)}}{\mathcal{Z}}\phi(\boldsymbol{x}\cdot\boldsymbol{w})\phi(\boldsymbol{z}\cdot\boldsymbol{w})d\boldsymbol{w} \quad (11)$$

Next, we proceed by calculating the matrix element of $\hat{Q}$. As shown in the integral appendix A.4, we have

$$\int d\mu_{\boldsymbol{x}}\hat{H}_1(\boldsymbol{x})\phi(\boldsymbol{x}\cdot\boldsymbol{w}) = \frac{2(\boldsymbol{w}^*\cdot\boldsymbol{w})}{\sqrt{\pi}\sqrt{\left(1+2|\boldsymbol{w}|^2\right)}}$$

$$\int d\mu_{\boldsymbol{x}}\hat{H}_3(\boldsymbol{x})\phi(\boldsymbol{x}\cdot\boldsymbol{w}) = -\frac{4(\boldsymbol{w}^*\cdot\boldsymbol{w})^3}{\sqrt{6\pi}\sqrt{\left(1+2|\boldsymbol{w}|^2\right)^3}}$$

Since the fluctuations in the directions orthogonal to $\boldsymbol{w}^*$ are small, the quantity $|\boldsymbol{w}|^2$ appearing in the denominator can be approximated to $(\boldsymbol{w}\cdot\boldsymbol{w}^*)^2 + (d-1)\frac{\sigma_w^2}{d}$ as explained in the main text. Since we are taking large values of $d$, then $\frac{d-1}{d} \cong 1$. Additionally, we denote $q = \boldsymbol{w}\cdot\boldsymbol{w}^*$, so that .

$$\tilde{Q} = \frac{\sigma_a^2}{\mathcal{Z}}\begin{bmatrix} \int\frac{2q^2e^{-\mathcal{S}(q;b,c)}}{\pi(1+2(q^2+\sigma_w^2))}dq & -\int\frac{8q^4e^{-\mathcal{S}(q;b,c)}}{\sqrt{6}\pi(1+2(q^2+\sigma_w^2))^2}dq \\ -\int\frac{8q^4e^{-\mathcal{S}(q;b,c)}}{\sqrt{6}\pi(1+2(q^2+\sigma_w^2))^2}dq & \int\frac{8q^6e^{-\mathcal{S}(q;b,c)}}{3\pi(1+2(q^2+\sigma_w^2))^3}dq \end{bmatrix}$$

Where $\mathcal{Z} = \int dq e^{-\mathcal{S}(q;b,c)}$. These integrals are computed numerically, given, $b, c$ to obtain self-consistent equations. Thus, $\tilde{Q}$ is a function of $b, c$, and on the other hand we can use $\tilde{Q}$ to obtain an expression for $b, c$ through the GPR formula in this two-dimensional subspace,

$$\begin{bmatrix} b \\ \sqrt{6}c \end{bmatrix} = \bar{\boldsymbol{t}} = \left[\tilde{Q} + \frac{\sigma^2}{n}I_2\right]^{-1}\boldsymbol{y} = \left[\tilde{Q} + \frac{\sigma^2}{n}I_2\right]^{-1}\begin{bmatrix} 1 \\ \sqrt{6}\epsilon \end{bmatrix} \quad (12)$$

Using a saddle point approximation, the action can be expanded near it's minima to a quadratic polynomial, in order to solve the integrals appearing in the matrix elements of $\tilde{Q}$. This results in two equations for $b, c$ which can be solved numerically.

## A.4    INTEGRAL CALCULATIONS FOR THE TEACHER-STUDENT MODEL

Here we compute the integral $\left\langle \left( \boldsymbol{w}^* \cdot \boldsymbol{x} \right)^{2n+1} \phi \left( \boldsymbol{w} \cdot \boldsymbol{x} \right) \right\rangle_{\boldsymbol{x}}$ where $\boldsymbol{x} \sim \mathcal{N} \left( 0, I_d \right)$. This integral appears both in the computation of the action and in the calculation of the matrix elements of $\tilde{Q}$. Denote-

$$I_n \equiv \frac{1}{Z} \int d\boldsymbol{x} e^{-\frac{1}{2}\boldsymbol{x}^2} \phi \left( \boldsymbol{w} \cdot \boldsymbol{x} \right) \left( \boldsymbol{w}^* \cdot \boldsymbol{x} \right)^{2n+1}. \tag{13}$$

Where $Z = (2\pi)^{d/2}$ is a normalizing factor. This integral is given by

$$\begin{aligned}
I_n &\equiv \frac{1}{Z} \int d\boldsymbol{x} e^{-\frac{1}{2}\boldsymbol{x}^2} \phi \left( \boldsymbol{w} \cdot \boldsymbol{x} \right) \left( \boldsymbol{w}^* \cdot \boldsymbol{x} \right)^{2n+1} \\
&= \frac{2}{\sqrt{\pi}} \frac{\boldsymbol{w} \cdot \boldsymbol{w}^*}{\sqrt{\det \left( \boldsymbol{1} + 2\boldsymbol{w}\boldsymbol{w}^T \right)}} \left\langle \left( \boldsymbol{w}^* \cdot \boldsymbol{x} \right)^{2n} \right\rangle_{\mathcal{N}\left( 0, \left( \boldsymbol{1} + 2\boldsymbol{w}\boldsymbol{w}^T \right)^{-1} \right)} + 2n \left| \boldsymbol{w}^* \right|^2 I_{n-1} \\
&= \frac{2}{\sqrt{\pi}} \frac{\boldsymbol{w} \cdot \boldsymbol{w}^*}{\sqrt{1 + 2 \left| \boldsymbol{w} \right|^2}} \left\langle \left( \boldsymbol{w}^* \cdot \boldsymbol{x} \right)^{2n} \right\rangle_{\mathcal{N}\left( 0, \left( \boldsymbol{1} + 2\boldsymbol{w}\boldsymbol{w}^T \right)^{-1} \right)} + 2n \left| \boldsymbol{w}^* \right|^2 I_{n-1}
\end{aligned} \tag{14}$$

By Sherman Morrison's formula,

$$\left( \boldsymbol{1} + 2\boldsymbol{w}\boldsymbol{w}^T \right)^{-1} = \boldsymbol{1} - \frac{2\boldsymbol{w}\boldsymbol{w}^T}{1 + 2 \left| \boldsymbol{w} \right|^2}. \tag{15}$$

Therefore, a recursive expression for $I_n$ can be obtained in terms of $\boldsymbol{w}$ and $I_{n-1}$

$$\begin{aligned}
I_n &= \frac{1}{Z} \int d\boldsymbol{x} e^{-\frac{1}{2}\boldsymbol{x}^2} \phi \left( \boldsymbol{w} \cdot \boldsymbol{x} \right) \left( \boldsymbol{w}^* \cdot \boldsymbol{x} \right)^{2n+1} \\
&= \frac{2}{\sqrt{\pi}} \frac{\boldsymbol{w} \cdot \boldsymbol{w}^*}{\sqrt{1 + 2 \left| \boldsymbol{w} \right|^2}} \left\langle \left( \boldsymbol{w}^* \cdot \boldsymbol{x} \right)^{2n} \right\rangle_{\mathcal{N}\left( 0, \left( \boldsymbol{1} + 2\boldsymbol{w}\boldsymbol{w}^T \right)^{-1} \right)} + 2n \left| \boldsymbol{w}^* \right|^2 I_{n-1}.
\end{aligned} \tag{16}$$

The cases $n = 0, 1$ are the ones used in this work. For $n = 0$ the integral is simply-

$$I_0 = \frac{2}{\sqrt{\pi}} \frac{\boldsymbol{w} \cdot \boldsymbol{w}^*}{\sqrt{1 + 2 \left| \boldsymbol{w} \right|^2}}. \tag{17}$$

For $n = 1$ the calculation is slightly more complicated. We have

$$\begin{aligned}
\left\langle \left( \boldsymbol{w}^* \cdot \boldsymbol{x} \right)^2 \right\rangle &= \sum_{ij} w_i' w_j' \left\langle x_i x_j \right\rangle = \sum_{ij} w_i' w_j' \left[ \left( \boldsymbol{1} + 2\boldsymbol{w}\boldsymbol{w}^T \right)^{-1} \right]_{ij} \\
&= \sum_{ij} w_i' w_j' \left[ \delta_{ij} - \frac{2 w_i w_j}{1 + 2 \left| \boldsymbol{w} \right|^2} \right] \\
&= \left| \boldsymbol{w}^* \right|^2 - \frac{2 \left( \boldsymbol{w} \cdot \boldsymbol{w}^* \right)^2}{1 + 2 \left| \boldsymbol{w} \right|^2}.
\end{aligned} \tag{18}$$

Which yield,

$$I_1 = \frac{2}{\sqrt{\pi}} \frac{\left( \boldsymbol{w} \cdot \boldsymbol{w}^* \right)}{\sqrt{\left( 1 + 2 \left| \boldsymbol{w} \right|^2 \right)}} \left( 3 \left| \boldsymbol{w}^* \right|^2 - \frac{2 \left( \boldsymbol{w} \cdot \boldsymbol{w}^* \right)^2}{1 + 2 \left| \boldsymbol{w} \right|^2} \right) \tag{19}$$

# B  MODULAR ALGEBRA

## B.1  MODULAR ALGEBRA KERNEL AND SYMMETRIES

We are first concerned with the spectral properties of $Q_{nm,n'm'}$, the latter being the average of $\tilde{Q}_{nm,n'm'}$ with respect $S[\boldsymbol{w}]$. To this end note some useful model symmetries that occur when $n_{\text{train}}$ spans the entire training set. By symmetries here we mean that the training algorithm, neural network, and the data-set are invariant remain invariant under the following transformations:

**I.** Taking $[n,m] \to [(n+q) \bmod P, (m+q') \bmod P]$, and $f_p \to f_{(p+q+q') \bmod P}$ with $q, q' \in \mathbb{Z}_P$

**II.** Taking $[n,m] \to [qn \bmod P, qm \bmod P]$ and $f_p \to f_{qp \bmod P}$ for $q \in \mathbb{Z}_P$ but different than zero.

Notably, $Q$ does not contain any reference to a specific label ($p$). Hence, both these symmetries are symmetries of $Q$ (both before and after any potential phase transition). Specifically, let us denote by $T_1$ ($T_2$) the orthogonal transformation taking $[n,m] \to [n+1,m] \bmod P$ ($[n,m] \to [n,m+1] \bmod P$) and by $C_q$ the transformation taking $[n,m] \to [qn,qm] \bmod P$. Then the following commutation relations holds,

$$[Q,T_1] = [Q,T_2] = [Q,C_q] = 0 \qquad (20)$$
$$[T_1,T_2] = 0$$

This implies that $Q, T_1$ and $T_2$ can be diagonalized on the same basis. The Fourier basis ($\phi_{k,k'}$), discussed in the main text, clearly diagonalizes $T_1$ and $T_2$ simultaneously. Furthermore, as no two $\phi_{k,k'}$ share the same eigenvalues of $T_1$ and $T_2$, each $\phi_{k,k'}$ must be an eigenfunction of $Q$.

Next, we notice that since $[Q,C_q] = 0$, $C_q\phi_{k,k'}$ must also be an eigenvector of $Q$ with the same eigenvalue. However, since $[T_{1,2}, C_q] \neq 0$ $C_q$ shifts us between, $\phi_{k,k'}$ hence implying degenerates. Based on the definition of the Fourier modes, we find $C_q\phi_{k,k'} = \phi_{qk,qk'}$ hence all pairs of momentum $(k,k',g,g')$ such that $k = qg, k' = qg'$ must have degenerate eigenvalues. Notably for prime $P$ for any $k, g \neq 0$ there exists a $q$ such that $qk = g$. We furthermore note that an additional "flip" symmetry exists, implying that $k, k'$ are associated with the same eigenvalue as $k', k$.

The above results for $Q$ hold exactly regardless of network width, noise, or other hyperparameters. Next, we focus on $Q$ at the NNGP limit ($N \to \infty$), which according to our saddle-point treatment, is also the kernel at any point before the phase transition (at large $P$). Here we find an enhanced symmetry which explains its poor performance. Indeed, before the phase transition, The kernel $Q$ is given

$$Q_{nm,n'm'} = \sigma_a^2 \left[ K_{nm,nm} K_{n'm',n'm'} + 2K_{nm,n'm'}^2 \right] \qquad (21)$$
$$K_{nm,n'm'} = \langle (W_n + W_{m+P})(W_{n'} + W_{m'+P}) \rangle_{\boldsymbol{w} \sim N[0, I_{2P \times 2P}]}$$
$$= \delta_{n,n'} + \delta_{m+P,m'+P} + \delta_{n,m'+P} + \delta_{m+P,n'}$$
$$= \delta_{n,n'} + \delta_{m,m'}$$

where we recall that $n, m \in [0...P-1]$. Given that, one can act with symmetry II separately on each index. This extra symmetry implies, via the above arguments, that all $k, k'$, and $q, q'$ with $k = ck'$ and $q = c'q'$ are equivalent. For prime, $P$ this results in only 3 distinct eigenvalues - the one associated with $k, k' = 0, 0$, the one associated with all $0, k \neq 0$ and $k \neq 0, 0$ modes and the one ($\lambda$) associated with $k \neq 0, k' \neq 0$. Notably, the target is supported only on this last $(P-1)^2$-dimensional degenerate subspace. The equal prior on $(P-1)^2$ eigenfunctions then implies $O(P^2)$ examples are needed for the GP to perform well on the task here, namely the entire multiplication table.

## B.2  FOURIER TRANSFORM OF THE ACTION

Here we compute the action in Fourier space without the weight decay term which appears in the main text. To construct the Fourier transformed action, we start from the fluctuating kernel term ($\tilde{Q}$)

given by

$$\sigma_a^{-2}\tilde{Q} = \sum_{n,n'=0}^{P} \sum_{m,m'=P}^{2P-1} (w_m + w_n)^2 (w_{m'} + w_{n'})^2 |n,m\rangle \langle n',m'| \tag{22}$$

where we have used physics-style bra-ket notation to denote the basis on which the operator $\tilde{Q}$ is presented. For instance, $\tilde{Q}_{n_0 m_0, n_0' m_0'}$ is given by acting with $\langle n_0, m_0|$ from the left and $|n_0', m_0'\rangle$ from the right and using $\langle n_0, m_0 | n_1, m_1\rangle = \delta_{n_0, n_1}\delta_{m_0, m_1}$.

Based on the symmetries of the problem we consider the following coordinate transformation

$$w_m \mapsto \sum_{k=0}^{P-1} w_k e^{-\frac{2\pi i}{P} kn}, \quad w_n \mapsto \sum_{k=0}^{P-1} v_k e^{-\frac{2\pi i}{P} km} \tag{23}$$

Substituting this in the equation for $\tilde{Q}$:

$$\tilde{Q} = \sum_{kk'qq'} \underbrace{\sum_{m,n} \left( w_k e^{-\frac{2\pi i}{P} kn} + v_k e^{-\frac{2\pi i}{P} km} \right) \left( w_{k'} e^{-\frac{2\pi i}{P} k'n} + v_{k'} e^{-\frac{2\pi i}{P} k'm} \right) |n,m\rangle}_{J} \tag{24}$$

$$\cdot \underbrace{\sum_{m',n'} \left( w_q e^{\frac{2\pi i}{P} qn'} + v_q e^{\frac{2\pi i}{P} qm'} \right) \left( w_{q'} e^{\frac{2\pi i}{P} q'n'} + v_{q'} e^{\frac{2\pi i}{P} q'm'} \right) \langle n',m'|}_{J'}$$

Where we can simplify further

$$J = \sum_{mn} \left( w_k w_{k'} e^{-\frac{2\pi i}{P}(k+k')n} + v_k v_{k'} e^{-\frac{2\pi i}{P}(k+k')m} + w_{k'} v_k e^{-\frac{2\pi i}{P}(k'n+km)} + w_k v_{k'} e^{-\frac{2\pi i}{P}(kn+k'm)} \right) |n,m\rangle \tag{25}$$

$$\equiv P \left( w_k w_{k'} |k+k', 0\rangle + v_k v_{k'} |0, k+k'\rangle + w_k v_{k'} |k, k'\rangle + w_{k'} v_k |k', k\rangle \right),$$

(which also serves as the definition of the Fourier transformed bra-ket notation) and similarly for the conjugated term:

$$J' = P \left( \langle -q, -q' | w_q v_{q'} + \langle -q', -q | w_{q'} v_q + \langle -(q+q'), 0 | w_q w_{q'} + \langle 0, -(q+q') | v_q v_{q'} \right) \tag{26}$$

We next turn to the quadratic term

$$\sum_{n=0}^{P-1} w_n^2 = \sum_{n=0}^{P-1} \sum_{k,k'=0}^{P-1} w_k w_{k'} e^{\frac{2\pi i}{P}(k+k')n} \tag{27}$$

$$= P \sum_{k=0}^{P-1} w_k w_{-k},$$

using $\sum_{n=0}^{P-1} e^{\frac{2\pi i}{P}(k+k')n} = P\delta_{k,-k}$. The quadratic term in the action (main text equation 3.2) $|w|^2$ is given by

$$|w|^2 = \sum_{n=0}^{P-1} w_n^2 + \sum_{m=P}^{2P-1} w_m^2 = P \left( \sum_{k=0}^{P-1} w_k w_{-k} + \sum_{k=0}^{P-1} v_k v_{-k} \right) \tag{28}$$

The target can be also expressed in terms of the Fourier basis, noting that the mode zero is zero for $P$ prime

$$y_{nm}^p = P^{-1} \sum_{k=1}^{P-1} e^{-i2\pi kp/P} e^{i2\pi k(n+m)/P} = \langle n,m| \sum_{q=1}^{P-1} e^{-2\pi iqp/P} |q,q\rangle \tag{29}$$

The remaining term in the action (main text equation 23) following the anzats $\sigma^2 t = ay$, and considering the whole dataset. the involves the kernel and is given by

$$\sum_{p,mn,m'n'} y_{nm}^p \overline{y}_{n'm'}^p \tilde{Q}_{nmn'm'} \tag{30}$$

$$= \sum_{p,mn,m'n'} \langle n, m| \sum_{q=1}^{P-1} e^{-2\pi i q p/P} |q, q\rangle \langle n', m'| \sum_{q'} e^{2\pi i q' p/P} |q', q'\rangle \tilde{Q}_{nmn'm'}$$

$$= \sum_{mn,m'n'} \sum_{q,q'=1}^{P-1} \langle n, m \mid q, q\rangle \langle n', m' \mid q', q'\rangle \tilde{Q}_{nn'mm'} \sum_p e^{-2\pi i (q'-q)p/P}$$

$$= P \sum_{mn,m'n'} \sum_{q,q'} \langle n, m \mid q, q\rangle \langle n', m' \mid q', q'\rangle \tilde{Q}_{nn'mm'} \delta_{q,q'}$$

$$= P \sum_q \langle q, q| \left[ \sum_{mn,m'n'} \tilde{Q}_{nn'mm'} |n, m\rangle \langle n', m'| \right] |q, q\rangle$$

$$= P \sum_{q=1}^{P-1} \langle q, q| \tilde{Q} |q, q\rangle$$

Note that the chosen basis is orthonormal, so that $\langle t \mid k\rangle = \delta_{k,t}$ and thus all terms with $|0\rangle$ vanish and, and we are left with-

$$P \sum_{t=1}^{P-1} \langle t, t| \tilde{Q} |t, t\rangle \tag{31}$$

$$= P^3 \sum_{kk'qq'} \sum_{t=1}^{P-1} \langle t, t| (w_k v_{k'} |k, k'\rangle + w_{k'} v_k |k', k\rangle)(w_q v_{q'} \langle -q, -q'| + w_{q'} v_q \langle -q', -q|) |t, t\rangle$$

$$= P^3 \sum_{kk'qq'} \sum_{t=1}^{P-1} (w_k v_{k'} \delta_{k,t} \delta_{k',t} + w_{k'} v_k \delta_{k,t} \delta_{k',t})(w_q v_{q'} \delta_{-q,t} \delta_{-q',t} + w_{q'} v_q \delta_{-q,t} \delta_{-q',t})$$

$$= 4P^3 \sum_{t=1}^{P-1} w_t v_t w_{-t} v_{-t}.$$

The action in Fourier space is then given by

$$\mathcal{S}[\hat{\boldsymbol{w}}] = P \left[ \left( \frac{1}{2} \sum_{k=0}^{P-1} w_k w_{-k} + \frac{1}{2} \sum_{k=0}^{P-1} v_k v_{-k} \right) - \frac{2\sigma_a^2 a^2 P^2}{N\sigma^4} \sum_{k=1}^{P-1} w_k w_{-k} v_k v_{-k} \right]. \tag{32}$$

Note that since the weights are all real, the Fourier transforms obey $\overline{w_k} = w_{-k}$, we can therefore rewrite $\mathcal{S}$ in terms of the parameters $|w_k|, |v_k|$

$$\mathcal{S}[\hat{\boldsymbol{w}}] = P \left[ \frac{1}{2} \sum_{k=0}^{P-1} \left( |w_k|^2 + |v_k|^2 \right) - \frac{2\sigma_a^2 a^2 P^2}{N\sigma^4} \sum_{k=1}^{P-1} |w_k|^2 |v_k|^2 \right]. \tag{33}$$

So that the action can be separated into independent actions $\mathcal{S}_k$ that each depend on different values of $k$

$$\mathcal{S}[\hat{\boldsymbol{w}}] = \sum_{k\neq 0}^{P-1} P \left[ \frac{1}{2} \left( |w_k|^2 + |v_k|^2 \right) - \frac{2\sigma_a^2 a^2 P^2}{N\sigma^4} |w_k|^2 |v_k|^2 \right] + \frac{1}{2} \left( |w_0|^2 + |v_0|^2 \right)$$

$$\equiv \mathcal{S}_0(|w_0|, |v_0|) + \sum_{k\neq 0}^{P-1} \mathcal{S}_k(|w_k|, |v_k|) \tag{34}$$

### B.3  Solving the equation for $a$

As done in the teacher-student model, obtaining a self-consistent equation for the discrepancy requires averaging the kernel with respect to the weights. Here, $\tilde{Q}$ is given by-

$$\tilde{Q} = \sum_{k,k',q,q'} \left( w_k w_{k'} \left| k+k',0\right\rangle + v_k v_{k'} \left|0,k+k'\right\rangle + w_k v_{k'} \left|k,k'\right\rangle + w_{k'} v_k \left|k',k\right\rangle \right) \cdot \tag{35}$$
$$\cdot \left( \left\langle -q,-q'\right| w_q v_{q'} + \left\langle -q',-q\right| w_{q'} v_q + \left\langle -(q+q'),0\right| w_q w_{q'} + \left\langle 0,-(q+q')\right| v_q v_{q'} \right)$$

Since the action in Fourier space depends only on the magnitude of the weights and not on the phase, we deduce that the phases are uniformly distributed on the unit circle and are independent of the magnitudes. There are multiple types of terms that appear in $\tilde{Q}$, and we will calculate each part individually:

1. For terms of the form- $w_k v_{k'} w_q v_{q'}$, we have- $\langle w_k v_{k'} w_q v_{q'} \rangle = \langle |w_k| |v_{k'}| |w_q| |v_{q'}| \rangle \left\langle e^{i\left(\phi_k + \phi_q + \theta_{k'} + \theta_{q'}\right)} \right\rangle$. Since the phases of $v,w$ are independents, then $\left\langle e^{i\left(\phi_k + \phi_q + \theta_{k'} + \theta_{q'}\right)} \right\rangle = \left\langle e^{i\left(\phi_k + \phi_q\right)} \right\rangle \left\langle e^{i\left(\theta_{k'} + \theta_{q'}\right)} \right\rangle$. If $q \neq \pm k$ then $\left\langle e^{i(\phi_k + \phi_q)} \right\rangle = \left\langle e^{i\phi_k} \right\rangle \left\langle e^{i\phi_q} \right\rangle$ and since the phases are uniformly distributed over $[0, 2\pi]$ these averages vanish and if $q = k$ then $\left\langle e^{i(\phi_k + \phi_q)} \right\rangle = \left\langle e^{2i\phi_k} \right\rangle$ which vanishes too, so that $\left\langle e^{i(\phi_k + \phi_q)} \right\rangle = \delta_{k,-q}$. Similarly, $\left\langle e^{i\left(\theta_{k'} + \theta_{q'}\right)} \right\rangle = \delta_{k',-q'}$. Thus, we have- $\langle w_k v_{k'} w_q v_{q'} \rangle = \delta_{k',-q'} \delta_{k,-q} \left\langle |w_k|^2 |v_{k'}|^2 \right\rangle$

2. For terms such as- $\langle w_k w_{k'} w_q v_{q'} \rangle$, by averaging over the phases of $v_{q'}$ the term vanishes, and similarly terms like $\langle w_k v_{k'} v_q v_{q'} \rangle$ vanish.

3. We are left with terms of the form- $\langle w_k w_{k'} w_q w_{q'} \rangle$, for this not to vanish, we require $\phi_k + \phi_q + \phi_{k'} + \phi_{q'} = 0$ so that either $k = -k'$ and $q = -q'$, $k = -q$ and $k' = -q'$ or $k = -q', k' = -q$. The same is true for $\langle v_k v_{k'} v_q v_{q'} \rangle$.

In conclusion, the average kernel is given by-

$$\sigma_a^{-2} Q = \sum_{k,k'=0}^{P-1} \left\langle |w_k|^2 |v_{k'}|^2 \right\rangle |k,k'\rangle \langle k,k'| \tag{36}$$

$$+ \sum_{k,k'=0}^{P-1} \left\langle |w_k|^2 |w_{k'}|^2 \right\rangle \left( 2|k+k',0\rangle\langle k+k',0| + |0,0\rangle\langle 0,0| \right)$$

$$+ \sum_{k,k'=0}^{P-1} \left\langle |v_k|^2 |v_{k'}|^2 \right\rangle \left( 2|0,k+k'\rangle\langle 0,k+k'| + |0,0\rangle\langle 0,0| \right)$$

Since the target has no $|0\rangle$ component, it is orthogonal to the vectors $\langle k+k',0|$, $\langle 0,k+k'|$ and $\langle 0,0|$ so that these terms don't contribute when $Q$ acts on the target. We are left with-

$$\sigma_a^{-2} Q \boldsymbol{y}^p = \sum_{t=1}^{P-1} e^{\frac{2\pi i}{P} pt} Q |t,t\rangle = \sum_{t=1}^{P-1} e^{\frac{2\pi i}{P} pt} \sum_{k,k'=0}^{P-1} \left\langle |w_k|^2 |v_{k'}|^2 \right\rangle |k,k'\rangle \delta_{t,k} \delta_{t,k'} \tag{37}$$

$$= \sum_{k=1}^{P-1} e^{\frac{2\pi i}{P} pk} \left\langle |w_k|^2 |v_k|^2 \right\rangle |k,k\rangle = \sigma_a^{-2} \lambda \boldsymbol{y}^P$$

Since for all $k \neq 0$ the actions are identical, the mean

$$\lambda = \sigma_a^2 \left\langle |w_k|^2 |v_k|^2 \right\rangle = \sigma_a^2 \frac{\int \left( |w_k|^2 |v_k|^2 \right) e^{-S_k(|w_k|,|v_k|;a)}}{\int e^{-S_k(|w_k|,|v_k|;a)}},$$

is a constant function of $a$ for every $k$. Thus, it can be deduced that the target in an eigenvector of $Q$, with an eigenvalue $\lambda$. A self-consistent equation for $a$ is obtained using the GPR formula, where-

$$a = -\frac{\sigma^2}{\lambda(a) + \sigma^2} \tag{38}$$

### B.4 CALCULATING THE EIGENVALUE OF THE TARGET

In the limit of $P \gg 1$, the quantity $\left\langle |w_k|^2 |v_k|^2 \right\rangle$ can be estimated using a saddle point approximation. The minimum of the action can be found by minimizing with respect to $|w_k|, |v_k|$-

$$0 = |w_k| - \frac{8\sigma_a^2 a^2 P}{N\sigma^4} |w_k| |v_k|^2 + \epsilon |w_k|^5 \tag{39}$$

$$= |w_k| \left( 1 - \frac{8\sigma_a^2 a^2 P}{N\sigma^4} |v_k|^2 + \epsilon |w_k|^4 \right)$$

$$0 = |v_k| - \frac{8\sigma_a^2 a^2 P}{N\sigma^4} |w_k|^2 |v_k| + \epsilon |v_k|^5 \tag{40}$$

$$= |v_k| \left( 1 - \frac{8\sigma_a^2 a^2 P}{N\sigma^4} |w_k|^2 + \epsilon |v_k|^4 \right)$$

If $|w_k|, |v_k|$ are both nonzero, then from symmetry we assume that $|w_k|^2 = |v_k|^2$, so that

$$|v_k|^2 = |w_k|^2 = \frac{1}{\gamma} \left( \frac{2\sigma_a^2 a^2 P^2}{N\sigma^4} \pm \sqrt{\left( \frac{2\sigma_a^2 a^2 P^2}{N\sigma^4} \right)^2 - \gamma} \right) := w_{\pm}^2$$

If $|w_k| = 0$ then either $|v_k| = 0$ or $(-\gamma)^{-\frac{1}{4}}$, but since we chose a positive $\gamma$ this is not possible. Similarly choosing $|v_k| = 0$ results in $|w_k| = 0$ so that either both are zero, or we have- $|v_k|^2 = |w_k|^2 = w_{\pm}^2$. Using saddle point approximation, we have for $k \neq 0$

$$P(|w_k|, |v_k|) = \frac{4e^{-S_k(w_+, w_+)} \delta(|w_k| - w_+) \delta(|v_k| - w_+) + \delta(|w_k|) \delta(|v_k|)}{1 + 4e^{-S_k(w_+, w_+)}} \tag{41}$$

In principle the leading order fluctuations near zero should appear in the probability distribution. However, this term can be discarded since the fluctuations in $w_k, v_k$ are of order $P^{-1}$ and so this contributes to $\lambda$ terms of order $P^{-2}$ so these are higher order corrections and can be ignored. With this discrete probability distribution, the matrix elements of $Q$ can be computed as follows

$$\left\langle |w_k|^2 |v_k|^2 \right\rangle = \frac{4w_+^4 e^{-\mathcal{S}_k(w_+, w_+)}}{1 + 4e^{-\mathcal{S}_k(w_+, w_+)}} \tag{42}$$

## C EXPERIMENTAL SETUP

Networks were trained for 360000 epochs, with lr $2 \cdot 10^{-3}$ with varying noise and as detailed in the graphs. The networks were trained on 50 different randomly generated datasets, with at least 4 different networks trained on each dataset. Parameters of the networks for the $\beta = 1$ case were given by-

$$N = 700, n = 3000, d = 150, \sigma_w^2 = 0.5, \tilde{\sigma}_a^2 = 8, \epsilon = -0.3 \tag{43}$$

with $\sigma^2 \in \{0.1, 0.15, 0.2, 0.25, 0.3\}$. This was repeated for $\beta = 1/8, 1/2, 2$ and in each case took $\alpha = \beta$. The error bars on the negative log posterior plots were calculated by dividing up the set of 200 trained networks into 5 subsets and computing the histogram of the weights in the $w^*$ direction in each subset. The error was obtained by dividing the std of the different histograms by the square root of the number of samples per ensemble. In the calculation for the average network output, the value for $b, c$ was computed as an average over the different datasets, and the error is given by the std divided by the number of ensembles.

## D  TIME DEPENDENT GROKKING

The equations appearing in section [3] rely on the equivalent kernel (EK) approximation, where the number of data points is taken to infinity, allowing the substitution of summations with integrals. This approximation captures the underlying feature learning aspect of Grokking, but phenomena resulting from differences between the train and the test data such as delayed generalization remain obscured. Here we provide numerical evidence that, away from the EK limit (i.e. for finite datasets and finite GPR noise) the phase transition behavior we predicted using EK persists and becomes accompanied by the expected generalization behavior of models which Grok.

To showcase this, we demonstrate experimentally in the teacher student setup, that for finite sized datasets the qualitative phenomena described in the EK limit hold as well. Namely, a first order phase transition into a GMFL phase is observed also for networks not in the EK limit. Moreover, in this case delayed generalization between the train and test data is observed during training.

Here we study the teacher-student setup, we train an ensemble of 50 networks for $7 \cdot 10^6$ epochs, each with:

$$n = 600, \ \ d = 150 \ \ \epsilon = -1.2 \tag{44}$$
$$\sigma^2 = 0.05, \ \ \sigma_a^2 = 0.011, \ \ \sigma_w^2 = 0.5$$

We drive a phase transition by reducing the width of the network, $N$, from $N = 2800$ to $N = 700$ as done in Gromov (2023). As the width of the network approaches infinity the network would not learn the cubic component as these require $n \propto d^3$ Cohen et al. (2021). Lowering the width of the network allows the network to learn the cubic component. We comment that opposed to the experiments shown in fig. (3), we do not use $\sigma^2$ to drive the phase transition, since taking vanishing values of $\sigma^2$ puts us in the EK limit, and for large values of $\sigma^2$ the network would fail to learn both train and test, artificially minimizing the difference between the two.

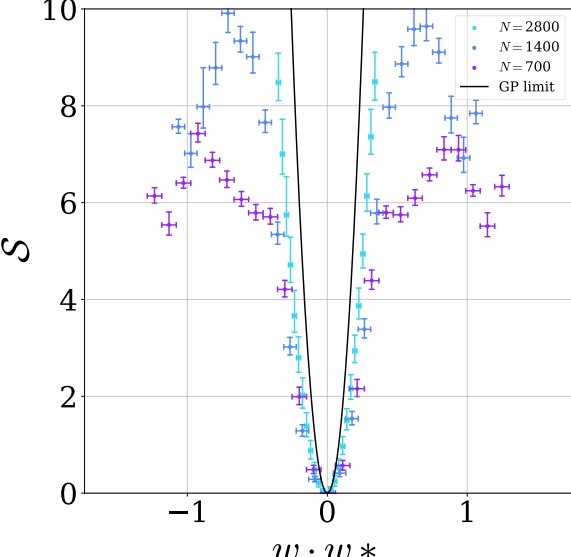

Figure 4: **Phase transition to GMFL-I phase for networks trained on finite sized data sets.** Here we use $N$ to drive the phase transition, and observe that for the smaller $N$s two new local minima appear in the action, which become more significant by decreasing the width.

Our experimental results are given in Figs. 4,5 and 6. Fig. 4 tracks the action $\mathcal{S}$, where $\mathcal{S} = -\log(p(\boldsymbol{w} \cdot \boldsymbol{w}_*))$ for different values of $N$: before the phase transition ($N = 2800$) shortly after the phase transition ($N = 1400$) and into the GMFL-I phase ($N = 700$). The error bars were computed by calculating the action for five subsets of the ensemble separately. The phase transition appearing here is also a first order phase transition, as demonstrated above for the EK limit.

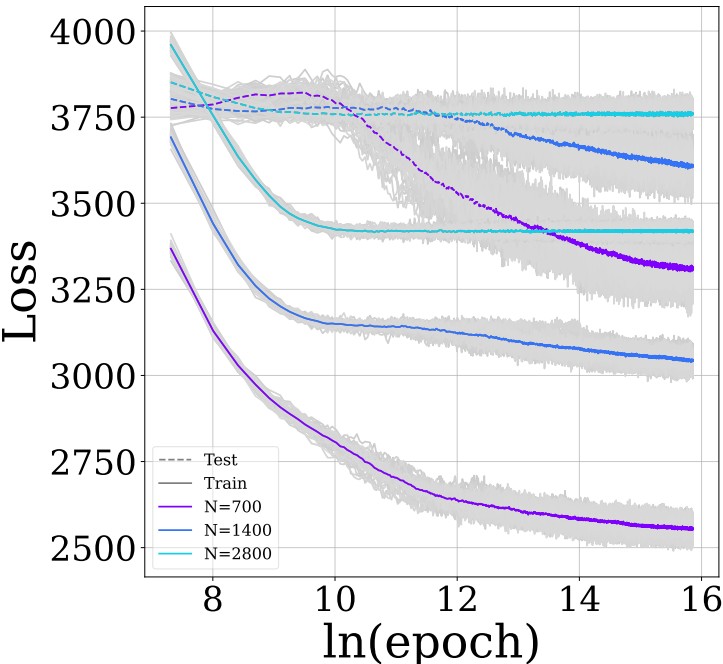

Figure 5: **Dynamic delayed generalization.** Here the train and test loss of an ensemble of 50 networks was tracked over the course of $7 \cdot 10^6$ epochs, for networks of three different width. The average loss is shown in color, and the individual networks in gray. At $N = 700$ a sharp drop in the test can be observed, as is commonly associated with Grokking Nanda et al. (2023); Gromov (2023).

Fig. 5 plots the train and test loss and Fig. 6 tracks the projection of the average network output on different normalized components of the target, as a function of the number of training steps for the different network widths and averaged over an ensemble of 50 networks. Where we define the linear component of the network to be- $H_1(\boldsymbol{w}^* \cdot \boldsymbol{x})$, and the cubic component as- $\epsilon H_3(\boldsymbol{w}^* \cdot \boldsymbol{x})$, and $\boldsymbol{x}$ is taken either from the test or the train dataset. Thus if the projection of the network on a certain normalized component is near one, it has been learned entirely by the network.

Both figures represent different ways in which there is delayed generalization. In Fig. 5, we observe the delayed generalization associated with Grokking in Nanda et al. (2023), where after an extended period of no change to the test loss or even a slight increase, a sharp drop appears. Another measure of generalization is improving how well the test and train compare in terms of learning output components. In Fig. 6, a delayed generalization is observed in the sense that the narrower networks (ie after the phase transition), continue learning the cubic component of the target for the test data, whereas the learning of the train data saturates and slows down.

In conclusion, we have demonstrated that feature learning phase transitions discussed in the main text are accompanied by delayed generalization associated with Grokking. The above result, together with the known Grokking behavior of the second model Gromov (2023), supports our conjecture that first-order phase transitions are a potential underlying mechanism of Grokking. We note that our formalism naturally extends to the finite size dataset regime by applying Cohen et al. (2021) to augment the EK results as also used in Refs. Naveh & Ringel (2021); Seroussi et al. (2023) in the context of kernel adaptation. Our focus on the EK limit in the analytical analysis was mainly for simplicity.

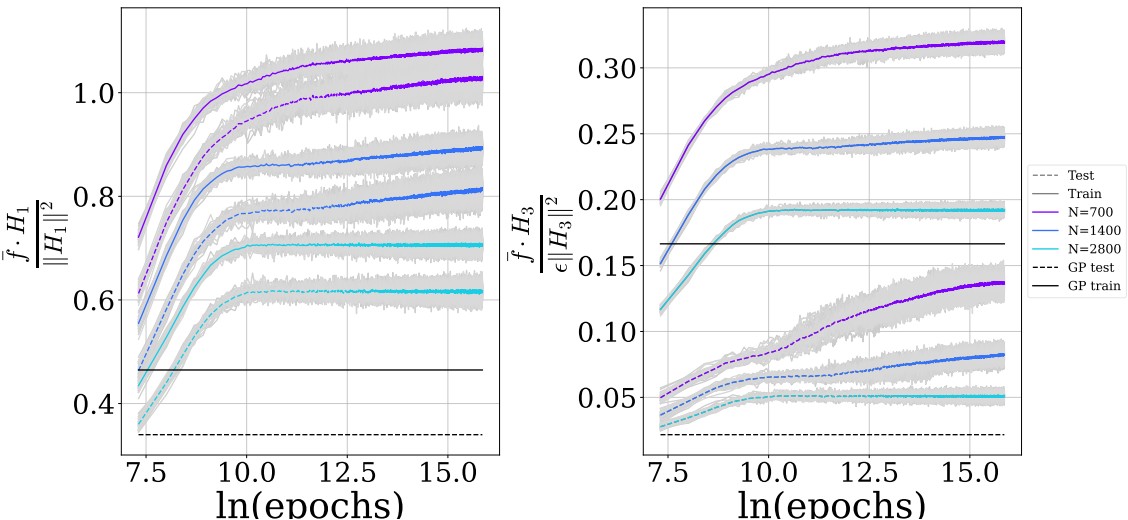

Figure 6: **Learnability of target components.** Here we track the output projection of network onto linear (lhs) and cubic (rhs) components of the target, both for test and train data, as a function of the ln of the epochs. The linear component of the target is given by the vector in data-space, whose components are given by- $H_1(\boldsymbol{w}^* \cdot \boldsymbol{x_\mu})$, with $\boldsymbol{x_\mu}$ drawn either from the train or the test. Similarly, the cubic component is given by- $\epsilon H_3(\boldsymbol{w}^* \cdot \boldsymbol{x_{mu}})$, where the $\epsilon$ factor is from the definition of the target. The average output as a function of either the train data or the test data can was be projected onto these normalized components. This computation was preformed on an ensemble of 50 networks, for networks of three different width. The average quantities are shown in color, and the individual networks in gray.

