# OpenReview forum: "Grokking as a First Order Phase Transition in Two Layer Networks"
_ICLR.cc/2024/Conference — ICLR 2024 poster_

### Official Review · Reviewer_iD3W · 2023-10-13

**Soundness:** 3 good
**Presentation:** 3 good
**Contribution:** 2 fair
**Rating:** 6
**Confidence:** 3

**Summary:**

This paper attributes grokking to a phase transition described by a Laudau-like theory using techniques of Gaussian processes. In the grokking process, representations form more and more "droplets", until these droplets come together to form water, completing the phase transition. The physical intuition is clear and makes sense. However, I'm not sure whether this theory really explains grokking or something else, given that there is no empirical experiments that compare NN training results to theory.

**Strengths:**

This paper is well-motivated and well written.

**Weaknesses:**

* The physical picture should be made clearer, to help non-physicist readers. For example, it would be nice to have a figure illustrating what you mean by "droplet" in terms of physics, and in terms of grokking.
* The link to NN training is unclear. How to translate the theory to NN training results?
* The fact that the analysis only applies to two-layer networks is not satisfactory but understandable.
* Missing some literature review on grokking, e.g., "Omnigrok: Grokking Beyond Algorithmic Data" and "Explaining grokking through circuit efficiency".

**Questions:**

See weaknesses above.

---

> ### Author Response · Authors · 2023-11-16
>
> We thank the reviewer for their precise summary, for pointing out the strengths and weaknesses of our work, and for the insightful and helpful comments we received from their review.
> These have considerably helped us to clarify the message. We address below each and every comment.
>
> # Weaknesses:
> * _The physical picture should be made clearer, to help non-physicist readers. For example, it would be nice to have a figure illustrating what you mean by "droplet" in terms of physics, and in terms of grokking._
>
> **Response:** Thank you for your suggestion (which is now included in the text). Note that following some similar comments from the first and fourth referees, the manuscript went through a major presentation overhaul. In particular, we delegated many of the model-specific derivation details to the appendix, provided a hopefully self-contained introduction to phase transitions, summarized the relevant parts of the kernel-adaptation work, and last but not least, separated our results from the derivation and presented them in a more concise way. We further provide much more compelling numerical evidence.
>
> In addition, following your suggestion we added Figure 1. which shows the phase diagram of the model illustrating the equivalence between the feature learning phases we find and gas water, and vapor phases.
> We sincerely hope that the referee will give the manuscript another fresh look.
>
> * _The link to NN training is unclear. How to translate the theory to NN training results?_
>
> **Response:** Our analysis provides predictions on the output of a network trained using full-batch gradient descent at a vanishing learning rate with white noise added to their gradients (see section 2.3 in the revised text). Our results are in very good agreement with numerical experiments on actual neural networks (see revised numerical Fig. 2).
>
> * _The fact that the analysis only applies to two-layer networks is not satisfactory but understandable._
>
> **Response:** The kernel adaptation formalism we use has been demonstrated to work well on deeper neural networks as well. Hence, we believe our approach, which is an extension of kernel-adaptation, can be extended to say three-layer networks at the price of tracking a few more parameters. However this would make the analytical solution significantly more complicated, and would obscure the general picture.
>
> * _Missing some literature review on grokking, e.g., "Omnigrok: Grokking Beyond Algorithmic Data" and "Explaining grokking through circuit efficiency"._
>
> **Response:** We agree and these references were added to the introduction.

---

> > ### Comment · Reviewer_iD3W · 2023-11-17
> > **Thanks for your response**
> >
> > Thanks for your response. One thing I'm still unsure is to what percentage this paper can explain the original grokking observed in Power et. al? I'm convinced that this is one of mechanisms responsible for grokking, but how significant is this? Said that, I think this is still an open question for grokking and it's unfair to ask the authors to address this question in this single paper. However, discussion on possible limitations may help (e.g., empirical results that your theory might not be able to explain). Overall I think authors addressed some of my concerns, so I'm raising my score to 6.

---

> > > ### Author Response · Authors · 2023-11-20
> > > **Relation to Power et. al.**
> > >
> > > We thank the reviewer for taking the time to reconsider our work.
> > > Regarding the relation to Power et. al. our theory for modular algebra does not currently take into consideration the fraction of data, but this can be included in future works, using EK type approximation we used for the teacher-student case augmented with noise renormalization of the type discussed in https://arxiv.org/abs/2002.02561 or https://arxiv.org/abs/1906.05301 . Making this extension and noting the similar data-fraction thresholds of ~0.2 are observed in fully-connected and transformers (https://arxiv.org/abs/2301.02679) -- we may hope to reproduce this 0.2 fraction ratio analytically.  Furthermore, a closely related approach to kernel adaptation is the DMFT approach which can capture dynamical effects. Using the symmetry arguments we derived one may hope to simplify the DMFT computation thereby perhaps obtaining the full dynamical picture.

---

### Official Review · Reviewer_64Ja · 2023-10-18

**Soundness:** 2 fair
**Presentation:** 4 excellent
**Contribution:** 3 good
**Rating:** 6
**Confidence:** 4

**Summary:**

The authors provide a heuristic analysis of two settings (a teacher-student model and a modular arithmetic model), which they show exhibit the grokking phenomenon. They relate this phenomenon to a first order phase transition, and provide a characterization of the effective action, with grokking corresponding to the transition from one minimum to the apparition of local minima.

**Strengths:**

The question addressed (grokking, and on a higher level feature learning) is an important one, and the authors provide an insightful and intriguing viewpoint  into how it arises. I also acknowledge the fact that the settings the authors set out to analyze are theoretically very challenging, making the provided insights valuable.

**Weaknesses:**

Overall, I find the presentation and clarity to be rather poor, making it hard to identify the claims and judge their soundness, and am in favour of rejection. If the authors clarify some points and promise to greatly overhaul the presentation I am happy to increase my score.

General remarks:
- The main presentation issue is to me the excessive focus on the technical details of the derivation,  from 2.4 to 3.1.2, at the expense of necessary details. For example, the discussion of the Langevin dynamics in 2.3 would strongly benefit from having the corresponding equation clearly written for definiteness, especially because several conventions exist, and also for the sake of readability.
- While I admit some technical details are essential in the understanding of the result and the flow of the paper,  this is not the case of all of them (e.g. the discussion at the beginning of 3.1.1 could be shortened).
- As a consequence, the assumptions on the scaling of $d,n,N$ are also dispersed through the text, making it difficult to follow which assumptions are needed in the end for the results to hold.

**Questions:**

- In 3.1, "$\sigma^2/n$ is kept fixed". Do you mean $\sigma^2 \sim n$ in scaling? Wouldn't the result of Cohen et al. rather hold for $\sigma^2=O(1)$?

- In 3.1.2, "thus the assumption that higher-order Hermite polynomials are irrelevant becomes inadequate." : I do not find which assumption this line points to. Does it refer to the ansatz (12)? Since the target only depends on $H_{1,3}$, I do not understand why the model would depend on higher order Hermites, nor why this would be a sign of feature learning. Could you elaborate?

- Am I right to understand the theory quantitatively predicts the experiments only in the GFL phase, and only holds qualitatively in terms of phenomenology to explain the transition to GMFL-I and II phases?

- (Minor) In Fig. 2, solid lines correspond to the theory? It should be explicitly stated for better readability.

- (Minor)  It would be more compelling to complement Fig. 3 with experimental simulations, similarly to Fig.1 middle and right.

---

> ### Author Response · Authors · 2023-11-16
>
> We thank thank the reviewer for their honest opinion, which has helped us restructure the work in a way we believe to be a significant improvement, as well as for their insightful questions. We think that following the reviewer's advice our message is significantly clearer. We address below every comment.
>
> # Weaknesses:
> * _Overall, I find the presentation and clarity to be rather poor, making it hard to identify the claims and judge their soundness, and am in favour of rejection. If the authors clarify some points and promise to greatly overhaul the presentation I am happy to increase my score._
>
> **Response:** We thank the reviewer for the time invested in our manuscript. We apologize for their poor reading experience. The manuscript now went through a major presentation overhaul. In particular, we delegated many of model-specific derivation details to the appendix, provided a hopefully self-contained introduction to phase transitions, summarized the relevant parts of the kernel-adaptation work, and last but not least, separated our results from the derivation and presented them in a more proof-like style. We further provide much more compelling numerical evidence.
>
> ## General remarks:
> * _The main presentation issue is to me the excessive focus on the technical details of the derivation, from 2.4 to 3.1.2, at the expense of necessary details. For example, the discussion of the Langevin dynamics in 2.3 would strongly benefit from having the corresponding equation clearly written for definiteness, especially because several conventions exist, and also for the sake of readability._
>
> **Response:** As aforementioned, we agree with the referee and made changes accordingly. In particular, the Langevin dynamics appear and are explained in further detail in the revised section 2.3.
>
> * _While I admit some technical details are essential in the understanding of the result and the flow of the paper, this is not the case of all of them (e.g. the discussion at the beginning of 3.1.1 could be shortened)._
>
> **Response:** We agree with the referee. These details can now be found in Appendix A.2,A.3.
>
> * _As a consequence, the assumptions on the scaling of $d,n,N$ are also dispersed through the text, making it difficult to follow which assumptions are needed in the end for the results to hold._
>
> **Response:** We again completely agree with the referee, and thank them for this comment. The derivation of the scaling limit now appears in App. A.1, whereas the statement of what scaling limit we take now appears at the beginning of our results section.

---

> ### Author Response · Authors · 2023-11-16
>
> # Questions:
>
> * _In 3.1, "is kept fixed" $\sigma^2 / n$. Do you mean $\sigma^2 \sim n$ in scaling? Wouldn't the result of Cohen et al. rather hold for  $\sigma^2  = O(1)$_
>
> **Response:** We agree with the referee that this is unclear, for a detailed explanation of the scaling here see App. A.1, in particular the $\alpha$ scaling. The results of this scaling also appear in the article itself. This does not contradict the conclusion in the referenced article as there to the EK approximation improves when $n,\sigma^2$ are increased simultaneously. More specifically, that work relies on Taylor expanding an interaction term of the form $n\int d\mu_x e^{-\sum_{p}(f_p(x)-y(x))^2/\sigma^2}$ to first order. This becomes even more justified as $\sigma^2$ grows larger.
>
> * _In 3.1.2, "thus the assumption that higher-order Hermite polynomials are irrelevant becomes inadequate." : I do not find which assumption this line points to... Could you elaborate?_
>
> **Response:** To clarify this, let us first consider the infinite width limit along with the large $n,\sigma^2$ limit. In this one needs to perform a spectral decomposition of the target function on the eigenbasis of the kernel. Doing so one finds that Hermit polynomials are (eigenvectors have eigenvalues which scale as $d^{poly-power}$, (up to negligible $d^{-poly-power}$ corrections). Hence as long as $n/\sigma^2$ scale as $d$ only linear functions are learnable (meaning that our $c$ is actually fixed to $\epsilon$). As our saddle point argument shows, before the phase transition the network is a slightly modified NNGP. However, this slight modification is not enough to overcome the $d^{poly-power}$ scaling of the eigenvalues or alter the fact the $H_{1/3}$ are eigenfunction. associated with the third Hermite polynomial. Hence the ansatz we use. The same holds just after the phase transition where however, the scaling with $d$ of the eigenvalue associated with $H_{3}$ changes like that of $H_1$. Going deeper into the droplet phase, the Gaussian mixture components become more probable, $H_{1/3}$ seize to be eigenfunctions of the kernel induced by the mixture state. Specifically, components like $Erf(w_* \cdot x)$ enter the eigenfunctions which we do not track. Hence even though the target is still a combination of the two Hermite polynomials the network's output would start having an $Erf(w_* \cdot x)$ like component. Tracking these latter effects quantitatively is left for future work.
>
> * _Am I right to understand the theory quantitatively predicts the experiments only in the GFL phase, and only holds qualitatively in terms of phenomenology to explain the transition to GMFL-I and II phases?_
>
> **Response:** The referee is right to question this issue. For the modular algebra case, our results are exact and hold throughout the three phases. However, for the teacher-student setting, new discrepancy modes, proportional to $Erf(w_* \cdot x)$, start growing just after the transition to GMFL-I. Thus we believe our results are exactly up to GFFL-I and just after the phase transition.
>
> Between submitting this work, and now, we conducted a more careful treatment of the scaling limit, which is now reflected in a much clearer match between theory and experiment.
>
> * _(Minor) In Fig. 2, solid lines correspond to the theory? It should be explicitly stated for better readability._
>
> **Response:** We replaced Fig. 2 by Fig. 2 (c) of the revised version, There are less lines and we hope that this is now clearer.
>
> * _(Minor) It would be more compelling to complement Fig. 3 with experimental simulations, similarly to Fig.1 middle and right._
>
> **Response:** We agree that this would be beneficial. However, in this setup we derived exact mathematical results and so did not see the need to further validate them. Experimental results can be added for camera ready version if necessary. Notwithstanding, we rely on the paper by Gromov in which the same qualitative behavior was shown experimentally.

---

> > ### Comment · Reviewer_64Ja · 2023-11-20
> > **Acknowledgement of rebuttal**
> >
> > I thank the authors for their detailed reply and the considerable rewriting of the paper, which is now much clearer. I particularly appreciate the addition of equation (11), which provides a welcome summary of the scaling limit considered. I have increased my score and am now in favour of acceptance.
> >
> > I find the discussion valuable, although a clear limitation of the work for the teacher/student setup is that the theoretical prediction only tightly holds in the regime where only the linear part of the teacher is learnable (please correct me if I am mistaken). As a side remark, there exist a related literature addressing learning single-index models with shallow nets which I believe would be beneficial for the authors to briefly connect to, e.g. [1].
> >
> > [1] Learning single index models with shallow neural networks, Bietti et al., 2022

---

> ### Author Response · Authors · 2023-11-22
>
> We thank the referee for this relevant reference. We now cite it along our discussion of possible sample complexity aspects of our results.
>
> Regarding the accuracy of the theory beyond the phase transition, we believe our approach captures the asymptotic behavior correctly before and exactly at the phase transition. In particular, we find that is also holds well  at least as far as $S(w)$ and $b$ are concerned, just after the transition. We believe this is because correction to our discrepancy ansatz ($t(x) = b H_1(x) + c H_3(x)$) have a parametrically small effect.  For instance taking $\sigma^2$ to be 0.015 smaller than our estimate $\sigma^2$ at the critical value (making $u$ larger than the critical value $u_c$), and allowing an additional  $d H_5(x)$ correction to $t(x)$ we found that $b,c,d = 0.2437, -0.2859, -0.002$ whereas $b,c = 0.2433,-0.2855$ without allowing for this correction. Consequently, we do not see this as a fundamental limitation but just a technical issue where one would have to identify more properly the relevant kernel eigenfunctions which span the target just after the transition. We thank the referee for fleshing out this point and clarify this further in the text.

---

### Official Review · Reviewer_MgBg · 2023-10-30

**Soundness:** 3 good
**Presentation:** 2 fair
**Contribution:** 4 excellent
**Rating:** 8
**Confidence:** 4

**Summary:**

This paper studies phase transitions in a model of deep neural networks. The authors focus on two distinct tasks: polynomial regression and modular arithmetic. Using an approach to study large width neural networks trained to equilibrium with Langevin dynamics, they derive an action that defines the Bayesian posterior distribution of read-in weight vectors given the training data. The authors show a competition of the Gaussian weight prior proportional to $\log p(w) \propto - \frac{1}{2} |w|^2$ and a data dependent likelihood term which arises from integration of the readout layer. Because of the competition of these terms, the action can transition from having one saddle point at $w=0$ (giving the lazy NNGP kernel + small feature learning corrections) to having multiple saddle points which all contribute to the final predictor. Beyond a certain point, the outlier overlap values dominate the action and the $w=0$ saddle point no longer contributes. These define the 3 phases of what the authors refer to as Grokking transitions and the three phases are Gaussian Feature Learning (GFL), then Gaussian Mixture Feature Learning GMFL-I and GMFL-II.  For tasks with a target function that is controlled by a single direction $w^*$ , they show that the overlap $w \cdot w^*$ has a distribution which becomes multimodal and that this can be made dramatic by scaling up the likelihood term. The authors argue that this phenomenon can lead to sudden changes in the learned representations and generalization error in neural networks as a function of $\sigma^2$ , dataset size $n$ or model width $N$ since all three of these quantities appear in the likelihood term, in analogy to first order phase transitions.

**Strengths:**

This paper extends a promising approach to neural network theory, known as the adaptive kernel approach, which studies how the kernels of deep networks adapt to data after feature learning. This paper provides two interesting case studies (polynomial regression and modular arithmetic) where they make progress on deriving an effective action which depends only on overlaps with the teacher direction $w^*$ or $v$. They show that the derived theory is accurate in simulations of networks on these learning tasks.

**Weaknesses:**

While the phenomena described and the resulting theoretical picture of 3 phases is quite impressive, I am not sure that this transition constitutes grokking as it is usually understood where training loss decreases much earlier than test loss during gradient based learning dynamics.  I do not see this as a fundamental limitation of the paper (which I quite appreciate) but mainly as an issue of framing. In my opinion this work is a more fundamental phenomenon than grokking since it pertains to fundamental questions in deep learning such as how width, data and parameterization affect feature learning. The paper is not completely rigorous and relies on various approximations, but I think that is completely fine since it supports its claims with experiments and computations at a physics level of rigor. It also relies on some prior work from Seroussi et al 2022 and a short summary of this approach in the Appendix could be helpful. Lastly, some of the exposition is a bit challenging to follow (see questions below).

That said, if these issues/questions are addressed I would be happy to raise my score.

**Questions:**

1. I am a bit confused about the scaling limits. What is the exact order of limits taken? Is width $N$ and data $n$ scaled together in some way? Is the kernel $Q$ always full rank? What about the scaling of $\sigma^2$ with various quantities? In the Appendix, this is discussed in a high level ($n$ large first, then $d$), but the resulting action still contains factors of $N, d, \sigma^2$ , etc rather than $O(1)$ quantities, which makes it a bit difficult to interpret. Is it thought that the $n$, $d$ limit commute? I suspect not, as the authors discuss various joint scalings like $n \sim d^3$ or $n \sim d^{1.5}$ etc in Appendix. Which (if any) of these are actually adopted when deriving the action?
2. In the Appendix A.1 the authors describe how they adopt a mean field parameterization, yet the resulting action’s likelihood term vanishes as $N \to \infty$. Is it clear why finite width networks would have less feature learning in the mean field parameterization? I was under the impression that this should be constant with $N$.
2. Could one study mean squared error so that the likelihood term was intensive in dataset size corresponding to a loss $\mathcal L = \frac{1}{n} \sum_{\mu=1}^n \ell(x_\mu, y_\mu)$ and parameter distirbution $p(\theta) \propto \exp\left(  - \beta \mathcal L - \frac{1}{2} |\theta|^2 \right)$? If this choice was made does the dataset size still control the phase transition? The reason I ask is because this would correspond to Langevin dynamics on a reasonably scaled and regularized cost function $\mathcal L + \frac{1}{2\beta} |\theta|^2$ with added Brownian motion with variance $\frac{2}{\beta}$, which at large $\beta$ is closer to how networks are trained in practice ( mean loss, rather than extensive sum over data points). I am wondering if many of the factors of $n$ which appear in the computation are in fact artifacts of the comparison of the raw scale of the likelihood to the prior.
4. I understand that the density of $w$ undergoes transitions as the number of critical points in the action changes, requiring use of multiple saddle points when computing observable averages. Why must a saddle point approximation over the density of weights $w$ be taken? One could also imagine computing observable averages by sampling this non-Gaussian density as in works on mean field neural networks like Mei et al 2019 or Bordelon & Pehlevan 2022? Is there something lost by approximating each ``well” around a minimum of the action $S(w)$ with a quadratic? Is this the approach taken by the authors when they predict macroscopic quantities like scale of cubic component in the predictor? Could the authors comment on these different approaches?
5. Could there be a connection between the phase transition reported here and the transition in https://arxiv.org/abs/2210.02157 Figure 5c where as the feature learning strength $\gamma_0$ increases, the training dynamics transition from having a convergent perturbation series in $\gamma_0^2$ to a non-perturbative regime where the power series diverges?

---

> ### Author Response · Authors · 2023-11-16
>
> We thank the reviewer for their precise summary, for pointing out the strengths and weaknesses of our work, and for the
> insightful and helpful comments we received from their review. We also appreciate the reviewer taking the time to ask relevant and thoughtful questions. The comments have helped us clarify our message and improve our work. We address below each and every comment.
>
> # Weaknesses:
> * _While the phenomena described and the resulting theoretical picture of 3 phases is quite impressive, I am not sure that this transition constitutes grokking... It also relies on some prior work from Seroussi et al 2022 and a short summary of this approach in the Appendix could be helpful. Lastly, some of the exposition is a bit challenging to follow (see questions below). That said, if these issues/questions are addressed I would be happy to raise my score.}_
>
> **Response:**
> We thank the referee for their insightful comments and are pleased to hear they find our work touches on fundamental issues.
> While we understand the viewpoint that Grokking is mainly about time, we wish to point out that (1) It is often accompanied by Grokking effects as a function of data-set size (See new Refs. in the introduction) (2) In general, behavior of DNNs as a function of time versus equilibrated neural networks as a function of dataset size is often similar, as reflected for instance in the use of One-Step SGD (You 2014 :"Exploring one pass learning for deep neural network training
> with averaged stochastic gradient descent"). We thus hope to keep this more timely framing, although we agree that our work addresses even more fundamental questions in learning representations.
>
> The revised manuscript now includes a more detailed description of the relevant parts of the adaptive kernel approach we are using (Sec. 2.4.1.). If the referee finds it useful, we can further expand on this in the appendix.

---

> ### Author Response · Authors · 2023-11-16
>
> # Questions:
> * _I am a bit confused about the scaling limits. What is the exact order of limits taken?...Which (if any) of these are actually adopted when deriving the action?_
>
> **Response:** Following the referee's comment, we clarified this point in the paper. In particular, we stress four conditions that our scaling limit should satisfy and find a two-dimensional family of scaling limits, parameterized by $\alpha,\beta$, which obey these conditions and yield the same performance as $\alpha,\beta$. The parameter $\beta$ embodies the large $N$ and large mean-field-scaling parameter limit together with the large input dimension limit. The parameter $\alpha$ embodies the large $n$ and GPR noise ($\sigma^2$) limit, making the equivalence approximation exact. Notably, apart from an overall scaling by $\sqrt{\beta}$ both $\alpha$ and $\beta$ do not change the action (see also Table 1 in the paper for the precise scaling for each model).
>
> * _In Appendix A.1 the authors describe how they adopt a mean field parameterization, yet the resulting action’s likelihood term vanishes as $N\to \infty$. Is it clear why finite-width networks would have less feature learning in the mean-field parameterization? I was under the impression that this should be constant with $N$._
>
> **Response:** Indeed, this is due to our notations being slightly confusing. The non-linear term in the action has indeed a negative power of $N$, but also a positive power of $\sigma_a^2$ and two negative powers of $\sigma^4$ both of which scale as the inverse mean-field scale. Had we taken just $\chi=N^{-1}$, while keeping $d$ and $n$ fixed, this would indeed give a constant action which is independent of $N$. However, in our scaling limit, we take a more moderate increase of $\chi$ (roughly as $\sqrt{N}$) and use the increase of $n$ to ensure the non-linear coefficient remains constant. This is all now clarified in detail in App. A.1, Eq. (2) in the revised appendix.
>
> * _Could one study mean squared error so that the likelihood term was intensive in dataset size, corresponding to a loss... I am wondering if many of the factors of $n$  which appear in the computation are in fact artifacts of the comparison of the raw scale of the likelihood to the prior._
>
> **Response:** The scaling we use is also related to the connection between the Langevin dynamics equilibrium distribution and the posterior distribution of Bayesian Neural Networks. Scaling the loss with a factor of $1/n$ implies in this setting that the variance of the noise scales with, $n$ i.e. $\beta^{-1} =2\sigma^{2}\sim n$ as well as the weight decay. Doing so one would find the exact same posterior and hence the same phenomenology.
>
> You are thus right that one could avoid this training-wise-unnatural scaling of the loss, at the price of being less natural from the point of view of Bayesian Neural Networks.
>
> * _I understand that the density of $w$ undergoes transitions as the number of critical points in the action changes, requiring use of multiple saddle points when computing observable averages... Could the authors comment on these different approaches?_
>
> **Response:** Change in the number of saddles of $S[w]$ is a prerequisite for a phase transition, however without some large-scale parameter scaling up $S[w]$ it would induce a crossover rather than a phase transition. In this crossover, the probability weight around one saddle gradually shifts to the other saddle instead of doing so abruptly. Thus while one can avoid taking a saddle point, the accuracy of the saddle point is in fact the reason for the phase transition. In practice, we calculated integral with these quantities both numerically and with the saddle point approximation and found negligible differences in the parameter regimes used in the experiments.
>
> * _Could there be a connection between the phase transition reported here and the transition in...?_
>
> **Response:**
> This is an interesting connection with DMFT and might imply potential dynamical aspects of our transition. Indeed, we can similarly view the mean-field scaling parameter ($\chi \sim \gamma_0$) as driving the phase transition. However, for this to be a phase transition, all other parameters must scale up. Both in the Fig. 5c of this work and in the text we did not find an emphasis on phase transitions but rather on transitions/crossovers. Perhaps the referee can clarify if they think an actual sharp transition is expected. Is this issue with convergence discussed there? Is it reflected in bifurcation points in the ODE for the alignment in Saxe's setting? We'd be thankful for some further clarifications so that we may cite this properly.

---

> > ### Comment · Reviewer_MgBg · 2023-11-18
> > **Response to Rebuttal**
> >
> > I really appreciate the detailed responses to my questions and significant updates the authors made to their paper. I am thus in favor of acceptance and will increase my score.
> >
> > On the last question, the movement of the kernels in the mean field linear network setting is $H - H_0 = \sqrt{1+\gamma_0^2}$ which has a power series expansion in $\gamma_0^2$ that converges only for $\gamma_0 < 1$. For $\gamma_0 > 1$ a perturbative expansion around the initial feature kernels will fail to capture the dynamics or final representations. This is not an equilibrium phenomenon so mapping it to a phase transition in the sense of Landau etc may not be accurate, but I was just curious if there was a connection.

---

> ### Author Response · Authors · 2023-11-20
> **Taylor expansions and phase transitions**
>
> We're happy to hear that the referee found our response useful.
>
> While we need to consider the referee's example further, we can say some related things.
>
> First, indeed the failure of a Taylor series to converge does not immediately mean a phase transition. For instance in quantum field theory, one often performs a perturbation theory keeping only 1-particle-irreducible diagrams leading to a mass renormalization (shifting of poles) in the Green's function $G(k,w)$. The resulting geometric series coming from perturbation theory is of the form $G=G_0 + G_0 V G_0 + G_0 V G_0 V G_0$ (where V is the self-energy) does not converge when $k,w$ are close to being on-shell and $V G_0$ is large. However, this does not imply a phase transition but rather just a limitation in this representation of the perturb Green's function. In practice one analytically continues the series. One way of doing so would be to re-sum this series leading to $1/ (G_0^{-1} - V)$ and treat this as the function. Another would be to find the Dyson equation obeyed by $G$ and solve it to again obtain this $1/(G_0^{-1} - V)$ form which has better analytical-continuation properties.
>
> Second, It would be fascinating to see these Landau-like phase transitions showing up in the DMFT and obtain their dynamical signatures. We wonder though if those can happen for deep linear networks. At least in equilibrium, we found that having a non-linear activation function is essential to get out of the GFL phase.
>
> We again thank the referee for his/her questions.

---

### Official Review · Reviewer_PXdZ · 2023-11-01

**Soundness:** 3 good
**Presentation:** 3 good
**Contribution:** 2 fair
**Rating:** 6
**Confidence:** 4

**Summary:**

The paper studies Grokking by building on prior work on adaptive kernel approaches for feature learning and claims that Grokking can be understood as a phase transition between different internal representations of the network. The authors explore feature learning in the two models studied (cubic teacher and modular addition) through this mapping to phase transitions.

**Strengths:**

- The formal treatment of Grokking using the approach proposed appears to be novel and compelling.
- Understanding the phenomenon through the lens of feature vs. lazy learning and casting it in the language of phase transitions is a direction of research that many prior works speculated about. This paper presents much interesting work in this direction.

**Weaknesses:**

- The connection to Grokking, as observed in prior work across various tasks and modalities, seems almost secondary in this paper. This work focuses on some toy models, solves them, and refers very little to the general phenomenon of delayed generalization.

The general feeling I have from this paper is that the approach somewhat obscures the contributions and implications. The formalism developed here and in the referenced work seems compelling, yet the paper does not go beyond two simple toy models. It’s not immediately clear which aspects can be generalized to other problems and what insights a reader can take away to their particular settings. It would have been great had the paper explored some of the directions suggested in the discussion section. For instance, providing a measure for Grokking on problems in the wild or studying whether using this formalism for pruning/regularization could be a fruitful application. Without these directions, I find it difficult to recommend a strong acceptance. But overall, I like the approach and would want to see more work in this direction, so I give a score of 6 (weak accept).

**Questions:**

- “We set the gradient noise level … under the equilibrium ensemble of fully-trained networks.” Feels ambiguous. Could you clarify?
- Could you clarify Figure 2? Specifically, the caption implies a qualitative change at the phase transition point, but I’m not sure what I’m supposed to be looking for. Also, how is it that increasing the strength (in abs terms) seemingly improves the linear component?
- Why is there a -1/P term in the definition of the target labels in the modular addition case?
- Is there a reason why the modular addition task is referred to as a student-teacher setting? It doesn’t feel like a teacher is involved here; the targets are simply the modular addition labels.

**Nits:**
- Typo: In section 3.1.2, “effecting amount of data” should be “effective amount of data”
- Better flow if Figures 1 and 2 were moved closer to the text explaining them. Otherwise, the reader might have to keep moving between pages.
- I think the paper could have a broader reach if improved intuition is provided (e.g., explaining how the terms in the action stem from the loss, regularization, and added noise more explicitly).

---

> ### Author Response · Authors · 2023-11-16
>
> We thank the reviewer for recognizing the novelty in our work and for the
> insightful and helpful comments we received from their review, as well as pointing out relevant further directions.
> These have considerably helped us to clarify the message. We also appreciate the attention to detail in the "Nits" section. We address below each question as well as every comment.
>
> # Weaknesses
> * _The connection to Grokking, as observed in prior work across various tasks and modalities, seems almost secondary in this paper. This work focuses on some toy models, solves them, and refers very little to the general phenomenon of delayed generalization._
>
> **Response:** Our work is indeed about learning representations, a topic we found fitting for ICLR. We understand the referee's viewpoint that framing this as a Grokking phenomenon only ignores the dynamic aspects of Grokking. Still, as the revised introduction now states and cites, Grokking, in the delayed generalization sense, is accompanied by Grokking as a function of dataset sizes. More generally, similar behavior is often observed between fully trained networks as a function of dataset size and networks trained on an infinite amount of data for a finite time (See for instance You 2014 :"Exploring one pass learning for deep neural network training
> with averaged stochastic gradient descent"). For instance, by taking very low learning rates and gradually increasing the data-set size, this correspondence between dataset size and time becomes linear and exact.
>
>    With regard to the referee's comment that we just solved two toy models, we note that we put out a mapping between abrupt feature learning effects and the theory of phase transitions. Our starting point action (log posterior distribution) is completely dataset and targets generic, and thus potentially captures a much bigger family of phase transitions in two-layer non-linear networks. We can solve it analytically for common toy models, like the ones we choose. Notably, one of which is a very natural and widely used toy model for Grokking (modular algebra learned by an FCN is also presented in other works such as Gromov 2023: "Grokking modular arithmetic" and in Nanda 2023: "Progress measures for grokking via mechanistic interpretability" ) hence hopefully elevating doubts that we somehow tailored the toy model to the theory.
>
>    Last we note that to the best of our knowledge and according to some of the other referee reports, solving a non-linear neural network with two trainable layers in the feature learning regime, is a considerable push to the theoretical state of the art in the field.
>
> * _The general feeling I have from this paper is that the approach somewhat obscures the contributions and implications... But overall, I like the approach and would want to see more work in this direction, so I give a score of 6 (weak accept)._
>
> **Response:** We sympathize with the referee's wish to see more practical implications. The current manuscript has been revised to stress our claims and the conditions under which they apply more clearly. It now places more places more emphasis on the generality of our approach by structuring the results sections so that it is evident that the same procedure is preformed in both cases and can be preformed similarly in many other similar cases. Furthermore, we extended the background sections thus hopefully further helping readers understand the potential generality of our approach.
>
> Turning to the second point regarding experiments, establishing a correspondence between feature learning in latent representation and water-vapor transitions, providing analytical pen & paper predictions on two canonical non-linear models, and verifying this experimentally is in our minds a sizable theoretical contribution. Moreover, as we now clarified in the text, our solution method (initial action and set of approximations we apply) is completely general and in particular shared between both models (see the derivation overview section 2.4). Furthermore, the droplet pictures, wherein some neurons are largely target agnostic whereas some carry important representations, can be used to rationalize why pruning works. Following this, we are currently conducting pruning experiments, showing some promising preliminary results.

---

> > ### Author Response · Authors · 2023-11-16
> >
> > # Questions
> > * _ “We set the gradient noise level … under the equilibrium ensemble of fully-trained networks.” Feels ambiguous. Could you clarify?_
> >
> > **Response:** We clarified this in the revised version, and hope that this is more clear now. We train using Gradient decent plus noise. We consider ensembles of many initializations and perform the training algorithm on each one of them. We set the noise so that it matches the GPR description, and there is a certain degree of freedom in the choice of the normalization of the weight decay term that is related to the freedom in the scaling of noise with $n$. We detail specific choices of parameters in the revised version.
> >
> >  * _Could you clarify Figure 2?..._
> >
> > **Response:** We replaced Figure 2, a similar version appears in Fig. 2  (c) of the revised version, since we agreed that it was unclear. We also explain this new graph in the main text.
> >
> > * _Why is there a -1/P term in the definition of the target labels in the modular addition case?_
> >
> > **Response:** Adding the 1/P makes it easier to work with the Fourier basis and remove the zero modes. Alternatively, one could remove this factor at the price of tracking two modes of the predictions, the current one and a bias term. The bias term would be learned very quickly, thereby effectively leaving us in the current setting, but only making the algebra more cumbersome.
> >
> > * _Is there a reason why the modular addition task is referred to as a student-teacher setting? It doesn’t feel like a teacher is involved here; the targets are simply the modular addition labels._
> >
> > **Response:** This is simply a matter of terminology. Assuming a target model is considering a specific teacher model in the so-called "Teacher-student-setting''.
> >
> > ## Nits:
> >
> > * _Typo: In section 3.1.2, “effecting amount of data” should be “effective amount of data”_
> >
> > **Response:** Corrected.
> >
> > * _Better flow if Figures 1 and 2 were moved closer to the text explaining them. Otherwise, the reader might have to keep moving between pages._
> >
> > **Response:** Thank you for this comment. We shifted the graphs to appear in the appropriate place in the text.
> >
> > *  _I think the paper could have a broader reach if improved intuition is provided (e.g., explaining how the terms in the action stem from the loss, regularization, and added noise more explicitly)._
> >
> > **Response:** We appreciate this insight, and we added in section 2.3 of the revised manuscripts further explanations to clarify the relation of some of the parameters to Langevin dynamics.

---

> > > ### Comment · Reviewer_PXdZ · 2023-11-23
> > >
> > > Thank you for the detailed reply and for answering my questions. It would be great if you could highlight significant changes in the revision (with a different color, for instance). That would make it much easier to find and evaluate the updates.

---

> > > > ### Author Response · Authors · 2023-11-23
> > > >
> > > > Thank you for the suggestion. The newest version is now color coded.

---

### Official Review · Reviewer_t2DF · 2023-11-05

**Soundness:** 2 fair
**Presentation:** 1 poor
**Contribution:** 2 fair
**Rating:** 3
**Confidence:** 3

**Summary:**

This paper makes attempts to explain the grokking phenomenon in deep learning via an adaptive kernel approach (Seroussi et al., 2023). Two settings are studied: a student erf-network learning a single index teacher; and a two-layer net with quadratic activation learning modular addition.

This paper doesn't introduce the background knowledge and related work at all, making it a very hard task to understand what they are exactly doing.

**Strengths:**

* This paper seems to have made some efforts to explain grokking, though I cannot fully understand them.

**Weaknesses:**

* This paper is poorly written. It claims that it is using an "adaptive kernel approach" to explain grokking, but they never explains what this method is. I tried to read the previous works, but they are not easy to read for ML audience, either. I urge the authors to introduce the background better: What is the "adaptive kernel"? Why is it important to study "action"? Why does the approximation in the paragraph beginning with "Next," in Page 4 make sense? What is the theory of phase transition in physics? How are phase transitions connected to bifurcation and saddle point equations? The current version of the paper contains too many unexplained jargons. Even if I could have spent hours reading previous works to understand the background better, I don't believe the current version of the paper is ready for ML audience to read.
* If this paper focuses on theory, then it is better to organize the claims into theorems and lemmas. In the current version, there are no theorems and lemmas at all. Everything seems to be stated in a very informal way, making it very hard to check the correctness.
* In my current understanding, the whole "adaptive kernel approach" builds on top of the Langevin dynamics, where isotropic noise is injected into the weights. If there is no noise or the noise is not isotropic, then the theory in this paper may not hold anymore. However, in the paper that proposes grokking (Power et al., 2022), the noise should be from the random sampling of batches and thus can hardly be isotropic, but the grokking phenomenon can still be observed. The adaptive kernel theory in this paper cannot cover this case at all.

**Questions:**

I would like to ask the authors to give more introduction of the technical tools they are using and rephrase their results in terms of theorems and lemmas.

---

> ### Author Response · Authors · 2023-11-16
>
> We thank the reviewer for their precise summary, for pointing out the strengths and weaknesses of our work as well as the places in which our background information was lacking. These have considerably helped us to both clarify the message and appropriately introduce the relevant concepts. We address below each and every comment.
> # Weaknesses
> * _This paper is poorly written ... I don't believe the current version of the paper is ready for ML audience to read._
>
>    **Response:** We thank the referee for the time invested in our work. We apologize for their poor reading experience here. We agree that while tools from statistical physics are making their way into the field, they are not at the stage where they can be taken for granted. Furthermore, our results in the previous version were too tangled up with derivation details.
>
>    The revised manuscript thus went through a major presentation overhaul, and we believe it now answers all points brought up by the referee. In particular, it now has a more self-contained introduction to kernel-adaptation results and to the theory of phase transitions. Furthermore, our main claims and the limits under which they apply are now clearly stated.
>
>    In essence, the idea behind kernel adaptation methods, is to extend the known result about Neural network at infinite width to the finite width regime. Previous results in this direction have shown that the output of the network at the end of training behaves like a Gaussian process. The kernel describing this Gaussian process as opposed to the Gaussian process at the infinite width limit do depend on the training data, both features and labels. Another useful aspect of this approach is that layers' probability distributions decouple, allowing one to formally treat representation learning as layer effect.
>
>    The reason why we focus on the action as our main result is that the action is the log of the posterior distribution up to a constant factor. Hence it is just a formality which helps avoid tracking normalization factors and terms in the exponent. We extend more on this idea and introduce the concept in section 2.4 in the main text.
> * _If this paper focuses on theory, then it is better to organize the claims into theorems and lemmas..._
>
>    **Response:** We agree with the referee that the conditions under which we believe our theory works should be stated much more clearly. Accordingly, we revised the scaling limit appendix (App. A.1) and clarified this limit in the main text. We also now sate all our claims explicitly in the main text in a way we hope will make it easier for the referee to follow.
>
>    Unfortunately, we do not aspire to prove all the claims in this work. Rather, we provide a physics-style treatment of this system using the same level of rigorousness and common practices found in most theoretical physics publications. For instance, if we find a small parameter, we expand in it and assume that such expansion convergences quickly. We however do not bound the Taylor residue, which is often very complicated to do. To elevate various doubts that accompany such a paradigm, we always perform numerical experiments of our approximations with numerics. In particular, the revised manuscript shows a very good match with experiments.
>
>    Notwithstanding, some of our claims are exact. For instance, with regard to the modular algebra section, our setup there is sufficient symmetry in the system such that our results are indeed mathematically exact.
>
>    We note that being non-rigors is also the strength of the Physists' approach, as it allows one to make progress and introduce claims motivated by physicist intuition without focusing on the technical details. These ideas can sometimes be beyond the current mathematical knowledge. Physists' claims can be then introduced for mathematicians as open problems and conjectures that are interesting to prove mathematically and sometimes take years to be proven. Such a process has happened many times in science in the past, spin glass theory is a very good example of that.

---

> ### Author Response · Authors · 2023-11-16
>
> # Weaknesses- continued
> * _In my current understanding, the whole "adaptive kernel approach" ... The adaptive kernel theory in this paper cannot cover this case at all._
>
>    **Response:** The precise outcome of training a neural network surely depends on all training details. However, it is rarely the case in practice where SGD at one batch size fails whereas at a large batch, it does spectacularly well. Underlying the celebrated NTK for example, is a seemingly impractical full-batch SGD at a vanishing learning rate. We assume full batch SGD at a vanishing learning rate and, heuristically, replace the SGD colored/correlated noise with white noise. This allows us to make contact with Bayesian Inference formalism. We comment that the latter shares much resemblance with SGD as argued in various places in the literature (for instance in Ard Louis, "is SGD a Bayesian sampler well almost").
> We additionally note that the adaptive kernel approach can also be viewed from the Bayesian perspective (see Li & Somplolisky 2021, Welling, M. & Teh, Y. W 2011 and more), where Langevin noise can be viewed as noise on the target and in many cases it is reasonable to assume that it is small and uniform over samples.
>
>    Separating proofs from arguments and exact agreement with quantitative agreement, we believe, based on experience, that the adaptive kernel approach is a good heuristic for realistic networks despite its assumptions. Indeed, our work is in qualitative agreement with various aspects of actual Grokking happening in SGD-trained networks (Gromov 2023), in particular, the types of filters that are learned and the fact that only a fraction "droplets" of neurons have the desired Fourier-transform filters.
>
> # Questions
>
> * _I would like to ask the authors to give more introduction of the technical tools they are using and rephrase their results in terms of theorems and lemmas_
>
>    **Response:** As aforementioned, the revised manuscript now contains a self-contained introduction to phase transitions and kernel adaptations approach (section 2.4), as well as more precise claims regarding our main results. We further made efforts to better state the conditions for our theory to hold and its implications.

---

### Author Response · Authors · 2023-11-17

We thank all the referees for the time invested in our work and their insightful comments. Apart from the detailed answers below, we like to point out that we substantially revised the background material, added clearer theorem-like statements of our results, and added further numerical results to validate our approximations. We believe our paper is now far better, much thanks to the referee's thoughtful comments.

---

### Author Response · Authors · 2023-11-23
**Color coded revision**

We uploaded a new version of the paper in which the significant additions we made during the rebuttal period appear in red.

---

### Meta-Review · Area_Chair_EZwR · 2023-12-05

**Metareview:**

This paper presents interesting theoretical results based on an approximate analytical method that can describe learning in the feature learning regime. It is concerned with two models for data and a learning model for which tight theoretical analysis is challenging. This contribution is interesting to the community and is why I vote for acceptance.

The presentation of the paper has two problems that I would urge the authors to fix:
(1) Why "droplets" of good representation? Why "mixed" phase? These two concepts are related to phase transitions in systems embedded in low-dimensional spaces (typically d=3) where nucleation creates droplets of the metastable phase, and phase co-existence can be observed. The models studied in this paper are of a mean-field nature corresponding to high-dimensional geometries where the metastable region associated with a first-order phase transition does not present phase co-existence (hence is not mixed) not droplet of the metastable phase. I am afraid that colleagues in machine learning who are not familiar with the physics of phase transition will be misled to think wrongly about these concepts, and I would recommend the authors change the title and the text, removing the references to "droplets" and called the "mixed" phase rather the "spinodal" region as common in other papers describing 1st order phase transition in high-dimensional spaces.

(2) The relation to Grokking is unjustified and seems to be a clickbait. Grokking was described to be associated to the fact that training loss gets very small, and only after (either in the number of samples or in time) the test error gets low. The present paper says nothing about this interplay between the train and test loss. The present paper seems to define Grokking as any sharp drop in the test error, which does not seem to coincide with the original phenomenon. I would recommend tuning down the connection with Grokking and concentrating on the actual contributions that are interesting by themselves and do not need these vague connections. It could well be that the authors were originally motivated by the Grokking phenomenon and it is fine to present it as a motivation, but they should re-think how to present their main contributions and the title of the paper.

**Justification For Why Not Higher Score:**

The presentation issues surely do not make this a spotlight.

**Justification For Why Not Lower Score:**

It could be rejected and probably accepted to the next conference once the author adjust the presentation.

---

### Decision · Program_Chairs · 2024-01-16

Accept (poster)